# The Fab region of IgG impairs the internalization pathway of FcRn upon Fc engagement

Maximilian Brinkhaus [1,2,3], Erwin Pannecoucke [3,4,5], Elvera J. van der Kooi [1,2], Arthur E. H. Bentlage [1,2], Ninotska I. L. Derksen [6], Julie Andries [4,5], Bianca Balbino[3], Magdalena Sips[3], Peter Ulrichts[3], Peter Verheesen[3], Hans de Haard [3], Theo Rispens [6], Savvas N. Savvides [4,5] & Gestur Vidarsson [1,2] ✉

Binding to the neonatal Fc receptor (FcRn) extends serum half-life of IgG, and antagonizing this interaction is a promising therapeutic approach in IgG-mediated autoimmune diseases. Fc-MST-HN, designed for enhanced FcRn binding capacity, has not been evaluated in the context of a full-length anti-body, and the structural properties of the attached Fab regions might affect the FcRn-mediated intracellular trafficking pathway. Here we present a comprehensive comparative analysis of the IgG salvage pathway between two full-size IgG1 variants, containing wild type and MST-HN Fc fragments, and their Fc-only counterparts. We find no evidence of Fab-regions affecting FcRn binding in cell-free assays, however, cellular assays show impaired binding of full-size IgG to FcRn, which translates into improved intracellular FcRn occupancy and intracellular accumulation of Fc-MST-HN compared to full size IgG1-MST-HN. The crystal structure of Fc-MST-HN in complex with FcRn provides a plausible explanation why the Fab disrupts the interaction only in the context of membrane-associated FcRn. Importantly, we find that Fc-MST-HN outperforms full-size IgG1-MST-HN in reducing IgG levels in cynomolgus monkeys. Collectively, our findings identify the cellular membrane context as a critical factor in FcRn biology and therapeutic targeting.

Monoclonal antibodies (mAb) of the IgG subclass or fragments thereof have become important therapeutic tools over the past decades[1]. Structurally, an IgG consists of two fragments antigen binding (Fabs) and the functionally distinct hinge and fragment crystallizable (Fc) region. IgG molecules are recognized by a variety of IgG-Fc receptors, one of them being the neonatal Fc receptor (FcRn)[2,3].

FcRn is a mostly intracellularly expressed, membrane-associated receptor[4] that is best known for mediating the long half-life of IgG[4–10] as well as placental transport of IgG from mother to unborn[11,12]. Over the years, FcRn has been identified to (co-)mediate several other IgG immune complex (IC)-driven reactions, such as phagocytosis[13], Ag cross-presentation[14–16], immune cell activation, and autoimmunity[17,18].

The interaction between IgG and FcRn is strongly pH-dependent[19–21]. IgG does not bind to FcRn at physiological pH, as found in the bloodstream, whereas it exhibits strong binding at acidic pH as found in endosomal compartments. Being highly expressed in recycling endosomes, FcRn rescues IgG from lysosomal degradation, recycles it back to the cell surface[22–24], and thereby mediates the relatively long half-life of IgG[4–9,19].

The critical residues for this pH-dependent interaction are I253, H310, and H435 in the CH2–CH3 region of the IgG-Fc fragment. In

combination with a decreasing pH during endosome development, protonation of H310 and H435 occurs, allowing for a charge-dependent interaction between the positively charged H310 and H435 on the Fc region of IgG and the negatively charged interface residues of FcRn[19,20,25,26].

Antagonizing the interaction between IgG and FcRn is pursued as a therapeutic strategy for the treatment of IgG-mediated autoimmune diseases[27]. Various FcRn antagonists are currently in clinical development and exist in different formats[28–32]. One such approach is using antibodies with increased affinity for FcRn compared with the natural Fc-ligand, so-called Abdegs—antibodies that enhance IgG degradation[28,33]. One of them is an IgG1-Fc fragment bearing five amino acid (AA) substitutions (M252Y, S254T, T256E, H433K, N434F, or "MST-HN")[28,29]. IgG1-MST-HN exhibits increased binding to FcRn relative to IgG1-WT at both physiological and acidic pH, but retains the natural pH-dependent binding profile[28,34]. This allows it to antagonize the IgG–FcRn interaction, while it can still be recycled in an FcRn-dependent manner[29,34]. By blocking the IgG salvage pathway, serum IgG is selectively reduced in vivo, including pathogenic IgG[28,29,31,32,35]. The therapeutic applicability of this approach is being tested in various clinical trials in indications[27] such as chronic inflammatory demyelinating polyneuropathy (CIDP) and myasthenia gravis (MG)[36], primary immune thrombocytopenia (ITP)[37], and pemphigus vulgaris/foliaceus (PV/PF)[38,39].

Several groups have reported that the charge distributions and other non-specific interactions in the complementarity determining regions (CDR) of the Fab region, can impact FcRn binding[40–42]. This can significantly impact the half-life of antibodies[40–43]. Steric factors also play an important role in FcRn binding, as IgG1 binds FcRn with the Fab regions potentially directed towards the cellular membrane[25,44], forcing IgG into a conformation to accommodate for the Fab arms. The specific conformations of FcRn-bound IgG in the cellular context, during the different stages of transport, in a cellular membrane context is subject to ongoing discussion[4,45,46]. However, the impact of these steric considerations on the design of antibody-based drugs, that utilize the FcRn-mediated transport pathway for half-life extension, is an important factor for the maximization of their delivery and therapeutic efficacy.

Here, we test the hypothesis of the Fab region interfering with the FcRn-mediated trafficking of IgG. We compare the biochemical, structural, and in vivo properties of a full-size IgG1-MST-HN to its Fc-only counterpart. We find that the Fab region hinders the interaction of IgG with membrane-associated FcRn, negatively affecting FcRn cellular handling of IgG. These results provide critical insights into how Abdegs bind to FcRn and into the cellular uptake of IgG, subsequent intracellular FcRn occupancy, accumulation of IgG, and FcRn-blocking capacity. Collectively, our findings provide the rationale for why Fc-only fragments are functionally superior to full-size IgG1 molecules in blocking IgG salvage in vivo, and are poised to influence therapeutic strategies targeting FcRn.

## Results

### Full-size IgG1 binds FcRn indistinguishably from Fc-only IgG1
First, all antibodies and antibody fragments used in the SPR experiments were analyzed using SDS-PAGE and HPLC-SEC to confirm integrity and size. All molecules showed the expected band sizes or elution profiles (Supplementary Fig. 1a) and were at least 98% monomeric (Supplementary Fig. 1b). To investigate the influence of the Fab arms on FcRn binding, we employed the IBIS SPR platform with random coupling of IgG onto the sensor array. Monomeric fractions of human FcRn were then titrated at pH 7.4 and pH 6.0, a setup described to avoid concentration-dependent avidity effects[47]. As expected, human FcRn did not bind any of the WT variants at pH 7.4, whereas low but detectable binding was found for the MST-HN variants. In line with previously published data, no binding was observed for human FcRn to

IgG1-IHH, regardless of the pH assayed[48]. At pH 6.0, both IgG1-WT and -MST-HN variants exhibited binding to human FcRn, whereas the MST-HN variants showed stronger binding (Fig. 1a). A comparison of the $K_D$ values derived by performing an equilibrium analysis and fitting a Langmuir 1:1 binding model revealed significantly better binding of the MST-HN variants compared to the WT variants at pH 6.0. No significant differences were found comparing the IgG1-Fc variants with the anti-HEL IgG1s with identical Fc parts (Fig. 1b). The affinity plots used to calculate $K_D$ values can be found in Supplementary Fig. 2. Next, we asked whether the Fab arms could have an impact on the elution pH by FcRn-chromatography, as suggested previously[49]. Both IgG1-WT and IgG1-MST-HN eluted at comparable pH as their Fc-only counterparts with the MST-HN variants eluting at a higher pH (pH 8.7) than their wild-type counterparts (pH 7.0) (Fig. 1c).

### Structural basis of FcRn engagement by Fc-MST-HN
To gain insights into the binding mode of the Abdegs to FcRn, we determined the crystal structure of Fc-MST-HN, both alone and in complex with FcRn (Supplementary Table S1). In the absence of structural models of human Fc-WT in complex with FcRn, the crystallographic model of Fc-MST[50] was used to gain insights into the differential binding modality of Fc-MST-HN to FcRn (Fig. 2). Overall, the crystal structure of Fc-MST-HN in complex with FcRn:β2M superimposes very well to that of FcRn:β2M in complex with Fc-MST ($C_\alpha$ RMSD of 0.62 Å using 380 atoms) (Fig. 2A). In both complexes, the Fc is bound by FcRn in an 2:1 molar ratio and in an upside-down fashion (i.e., with the CH2 domains oriented towards the membrane-proximal region of FcRn). No significant conformational changes could be observed in the rotameric conformations of residues known to mediate electrostatic or hydrophobic interactions between Fc-MST and FcRn[50] (Fig. 2B, C). Interestingly, although Fc-MST-HN predominantly adopts a predisposed binding interface (Supplementary Fig. 3a, b), three conformations for the side chain of Y252 could be observed when no binding partner was present, which all collapse into a single rotamer upon binding of FcRn (Supplementary Fig. 3c). Thus, this observation indicates that Y252 is allowed to freely sample rotameric space in the absence of a binding partner.

Our structural analysis established that while Fc-MST-HN retains all interaction epitopes on FcRn compared to Fc-MST, it introduces a novel interaction site centered around F434 in Fc-MST-HN, which exploits a hitherto unaddressed hydrophobic cavity in FcRn. The engagement of this site expands the hydrophobic Fc-MST-HN:FcRn interface with only a minimal conformational adaptation of FcRn at positions D130 and L135 (Fig. 2D, E). However, in contrast to this clear role of F434, the electron density around K433 did not allow modeling of its side chain, indicating that K433 does not participate in the interaction with FcRn. The crystal structure of unbound Fc-MST-HN, in which two opposing conformations of K433 could be observed (Supplementary Fig. 3d), does not clarify the possible role for a lysine at this position. Finally, thermodynamic analysis of both interactions demonstrated that whilst the enthalpic component of the interactions remained remarkably similar, the higher affinity of Fc-MST-HN over Fc-MST[28,34,51] could be solely attributed to a decrease of the entropic penalty (Supplementary Fig. 4), consistent with the increase in the hydrophobicity of the interaction due to the engagement of F434. Taken together, these findings demonstrate that Fc-MST-HN binds FcRn using a similar architecture as described for Fc-MST, and only deploys a larger hydrophobic interface to bind FcRn with a higher affinity.

### Fc only binds membrane-associated FcRn more efficiently than full-size IgG1
As our crystallographic analysis confirmed that Fc-MST-HN engages FcRn in a mode that projecting the Fab regions towards the membrane, we explored the possibility that Fc-only fragments would only bind

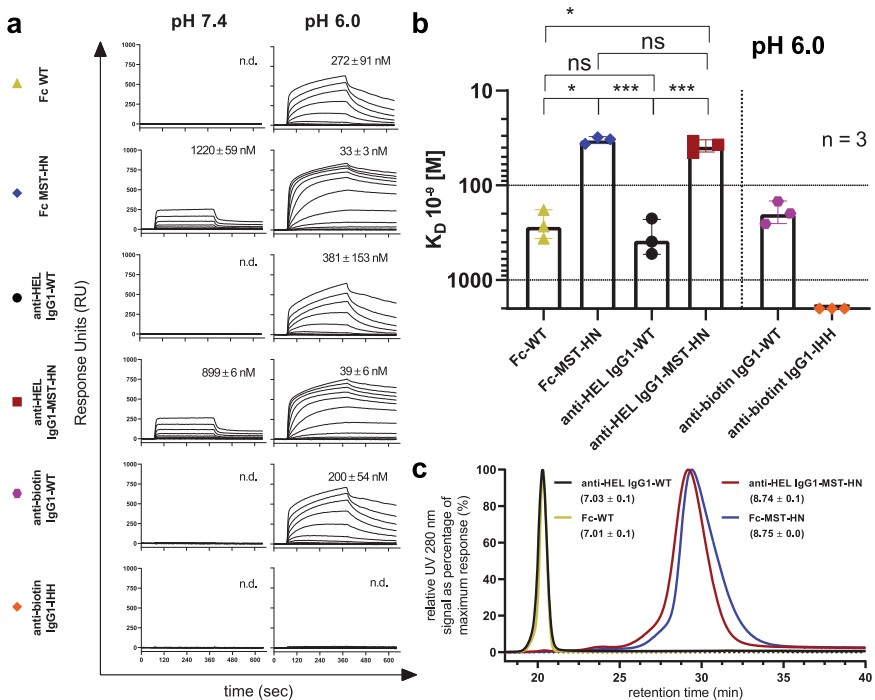

**Fig. 1 | No differences in FcRn affinity using SPR and elution pH using FcRn-affinity chromatography between full-size IgG1 and Fc only. a** Examples of sensorgrams of binding of IgG variants and indicated Fc fragments to FcRn as measured by SPR at pH 7.4 or 6.0. IgG variants were coupled to the SPR-array at a range of concentration (7.5, 15, 30, and 60 nM; shown is 60 nM for anti-biotin IHH and 15 nM for all other variants), with soluble FcRn injected from 0.49 to 1000 nM, represented by the different traces. The numbers indicate averaged mean $K_D$ values for three independent experiments, and the corresponding standard deviations in (nM) were calculated by performing an equilibrium analysis and fitting a 1:1 Langmuir binding model (Supplementary Fig. 2). "n.d." indicates that affinities were non-determinable. **b** Mean $K_D$ values of measurements at pH 6.0 as mentioned above shown in histograms. Error bars indicate standard deviations. **c** Chromatograms

obtained from comparing elution volume and elution pH (pH-gradient elution from pH 5.5 to pH 8.8) on a FcRn-affinity column for Fc-WT, Fc-MST-HN, anti-HEL IgG1-WT and anti-HEL IgG1-MST-HN using FcRn-affinity chromatography with relative UV 280 nm signal normalized to maximum response. Numbers indicate elution pH averaged from at least two independent experiments. Elution pH and chromatograms originate from independent experiments. Statistical analysis was performed by a two-sided Wald test, using a linear model with restricted maximum likelihood for the logarithm of the affinities with antibody as a fixed effect and incorporating heterogeneity of the residual variability for the different antibodies. *P* values were corrected by applying Hommel's procedure for multiple testing. Data in (**a**) are representative of three independent SPR runs, the results of which is summarized in (**b**). Statistical differences are indicated with asterisks: *<0.05, ***<0.001.

FcRn differently than full-size IgG in the context of a membrane-associated FcRn. We assessed binding of the Fc fragment and full-size IgG1 to cell-surface membrane-associated FcRn to more closely resemble the physiological context. For this purpose, a competition assay was developed to measure available FcRn at pH 6.0. In this assay, displacement of a directly labeled anti-FcRn Fab recognizing the IgG-binding interface and kept at a constant concentration, was monitored after coincubation with an FcRn inhibitor (WT or MST-HN) at increasing concentrations (Fig. 3a). The assay was performed on ice/ at 4 °C to prevent internalization at a range of molar ratios of anti-FcRn Fab:inhibitor (1:25–1:250 (Fig. 3b–d)). Binding of the anti-FcRn Fab to human FcRn at pH 6.0 and pH 7.4 was confirmed by SPR, showing binding with a 24.1 nM and 48.4 nM affinity at pH 6.0 and 7.4, respectively (Supplementary Fig. 5a). The general gating strategy for all flow cytometry-based experiments with HEK-FcRn-GFP cells is shown in Supplementary Fig. 5b. HEK cells overexpressing FcRn-GFP also express FcRn on the surface as seen by anti-FcRn Fab either without an inhibitor or IgG1-IHH (Fig. 3b). Both Fc-MST-HN and full-size anti-HEL IgG1-MST-HN almost completely outcompeted binding of the AF650-labeled anti-FcRn Fab at a concentration of 500 nM (Fig. 3b, c), suggesting binding to membrane-associated FcRn on the cell surface only. Furthermore, titration of the inhibitors led to a concentration-dependent shift of the anti-FcRn Fab-AF650 signal, confirming an overlapping epitope on FcRn with the MST-HN molecules. Equimolar treatment with Fc-MST-HN consistently blocked anti-FcRn Fab signal better than anti-HEL IgG1-MST-HN (Fig. 3b), which was found to be significant for all concentrations tested from 4 nM and higher. In line

with their lowered affinities to FcRn, WT IgG1 and Fc only significantly blocked binding of anti-FcRn Fab at the highest concentration, 500 nM (Fig. 3c), while the IgG1-IHH resulted in a signal comparable to medium control (Fig. 3b). At these high concentrations, the number of Fab arms on the IgG-Fc backbone (size and integrity verified by HPLC-SEC Supplementary Fig. 6) also affected the binding to membrane-associated FcRn (Fig. 3d), suggesting the Fab cargo negatively interferes with the interaction of the IgG-Fc.

## Fc-MST-HN shows higher occupancy of intracellular FcRn than full-size IgG1-MST-HN

Next, we asked if more efficient occupancy of cell-surface membrane-associated FcRn would also translate to elevated intracellular FcRn occupancy in intracellular compartments (e.g., endosomes, tubule-vesicular transport carriers)[22,52]. Therefore, we adapted the protocol initially published in ref. 32, but added extensive washing at pH 7.4 to wash away surface-bound IgG. In line with direct competition to surface FcRn on ice/ at 4 °C under acidic conditions (Fig. 3), no intracellular FcRn occupancy was found at 37 °C for the WT variants compared to IgG1-IHH or controls without IgG, whereas MST-HN molecules led to a concentration-dependent block of the anti-FcRn signal (Fig. 4a). In contrast to the results of the surface competition assay shown in Fig. 3b, c, loading the cells with the MST-HN molecules did not completely block binding of the labeled anti-FcRn Fab, likely due to unoccupied FcRn on the surface. However, also here, the Fc-MST-HN exhibited a significantly higher FcRn occupancy when compared to full-size anti-HEL IgG1-MST-HN (Fig. 4b). No difference was observed in

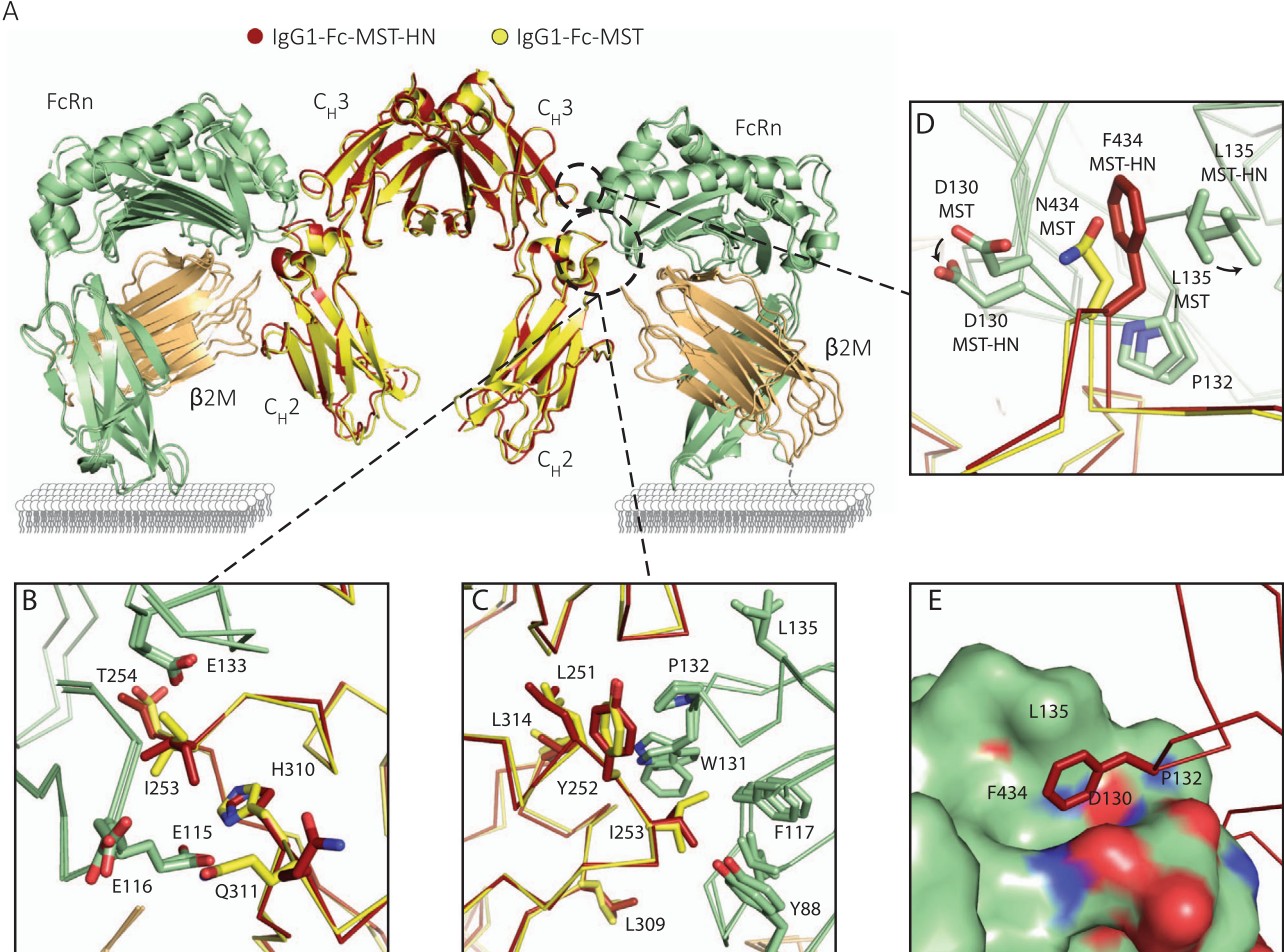

**Fig. 2 | Fc-MST-HN combines a larger hydrophobic-binding epitope on FcRn than on Fc-MST[50] with the expected upside-down mode of interaction.**
**A** Superposition of the crystallographic models of complexes between FcRn:ß2M (green and orange) and Fc-MST (yellow) or Fc-MST-HN (red), displayed in cartoon representation. The linker region connecting the extracellular region of FcRn to its transmembrane domain is represented as a dashed line. If present, Fab moieties would be oriented downwards and sterically clash with the membrane, which is shown for clarity. Detailed view of superimposed residues that contribute (**B**) electrostatically and (**C**) hydrophobically to the Fc:FcRn interface. Main chains are shown as ribbons, indicated side chains are shown as sticks. **D** Top view of the superimposed models, showing that accommodation of F434 in Fc-MST-HN (red) instead of N434 in Fc-MST (yellow) only requires a local conformational change by FcRn (green) at positions 130 and 135. Main chains shown as ribbons, indicated side chains are shown as sticks. **E** Side view of the superimposed models, detailing how F434 (red, stick representation) of Fc-MST-HN (red, ribbon representation) makes use of a hydrophobic pocket adjacent to L135 and P132. FcRn is shown in surface representation.

FcRn expression under these conditions, confirming that the differential treatment does not affect FcRn expression (Supplementary Fig. 7). To test if this observation is of a general nature rather than Fab specific, we tested two other full-size IgG1-MST-HN with different specificities. Receptor occupancy among all full-size IgG1 overlapped and was significantly different from Fc-MST-HN (Supplementary Fig. 8). We then investigated if these differences in intracellular FcRn occupancy might be influenced by differential internalization and intracellular accumulation of the FcRn inhibitors. Significantly more directly labeled Fc-MST-HN accumulated in the cells than full-size anti-HEL IgG1-MST-HN, whereas WT molecules remained barely detectable (Fig. 4c). Taken together, the results indicate that the IgG-Fab arms negatively affect the binding to membrane-associated FcRn. Furthermore, it may also lead to higher retention of the Fc-MST-HN in the cells than its full-size IgG counterpart.

### Fc-MST-HN is more efficient than full-size IgG1-MST-HN in blocking FcRn-dependent recycling of IgG1 in vitro
We then tested if the more efficient FcRn binding and accumulation in cells of Fc-MST-HN also translates to more efficient functional blocking of FcRn-mediated recycling. HEK-FcRn-GFP cells were loaded with our

FcRn-blocking reagents in the same manner that was used to evaluate intracellular FcRn occupancy and accumulation (Fig. 4), prior to uptake and recycling of anti-biotin IgG1 (Fig. 5). Without inhibitors, a clear FcRn dependency was observed, with only negligible amounts of IgG1-IHH being recycled (Fig. 5a). This FcRn-dependent recycling of anti-biotin IgG1-was blocked in a concentration-dependent manner by both MST-HN molecules, whereas the WT-IgG1 or -Fc fragments showed a trend (not significant) towards blockade when used at 500 nM. In line with the results described above, Fc-MST-HN antagonized FcRn-dependent recycling more effectively than full-size IgG1-MST-HN (Fig. 5b).

### Fc-MST-HN is more efficient than full-size IgG1-MST-HN in blocking FcRn-dependent recycling of IgG1 in cynomolgus monkeys
We then compared the capacity of both Fc-MST-HN and IgG1-MST-HN to block FcRn in vivo using an established model in cynomolgus monkeys[29]. In line with cellular binding and blocking of FcRn in cells, Fc-MST-HN reduced tracer antibody levels significantly more efficiently than IgG1-MST-HN or PBS control, when normalized to predose. The joint hypothesis test confirmed the effect of treatment compared

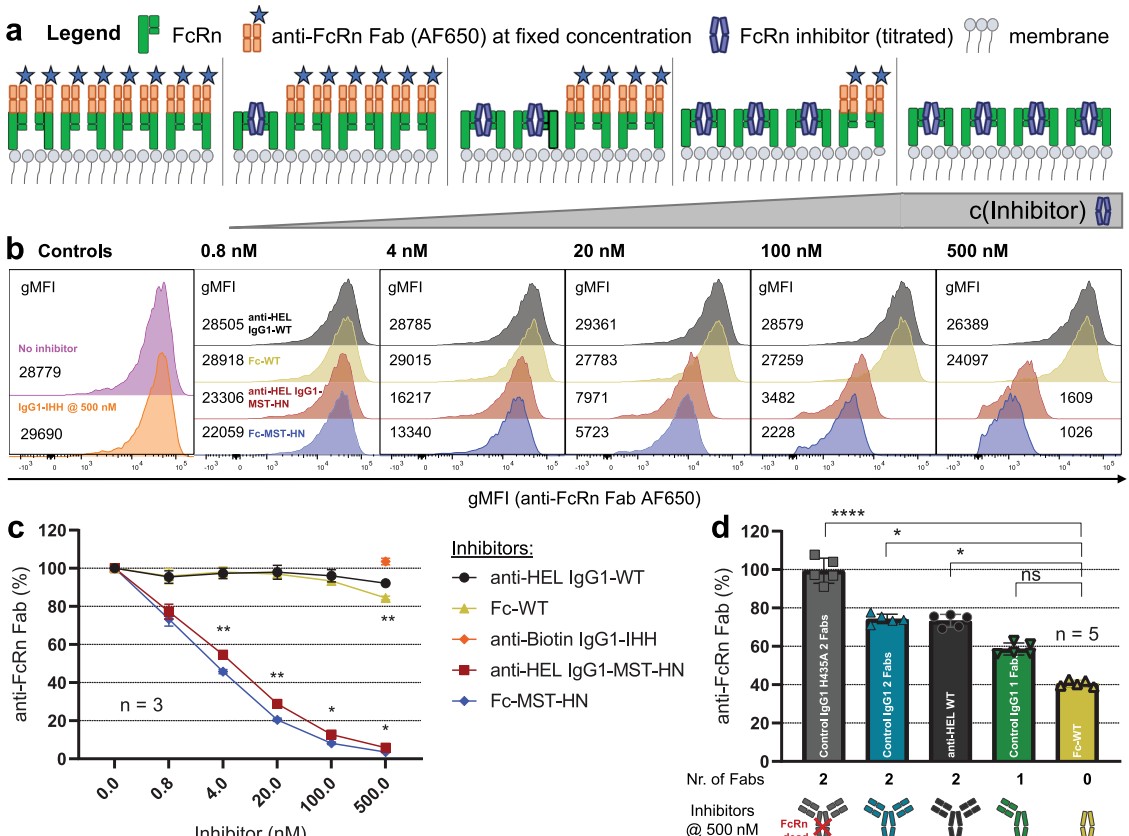

**Fig. 3 | Binding of IgG-Fab regions to membrane-associated FcRn and cellular uptake of IgG1. a** Schematic overview illustrating the principle of the surface competition assay to measure membrane-associated FcRn binding. **b** Histograms of a representative experiment comparing the binding of AF650-labeled anti-FcRn Fab to membrane-associated FcRn expressed on the surface of HEK-FcRn-GFP cells when co-incubated with the indicated FcRn inhibitors (full-size IgG1 or Fc fragments) at pH 6.0 on ice at 4 °C. Detection (AF650-labeled anti-FcRn Fab):inhibitor molar ratio at highest inhibitor concentration = 1:25 (optimized to measure competition with MST-HN containing IgG1). **c** Anti-FcRn Fab-AF650 staining of cells plotted as percent gMFI in the presence of indicated inhibitors compared with untreated cells. **d** Anti-FcRn Fab-AF650 staining of cells plotted as in (**c**), but for WT IgG1 variants of different specificities with indicated numbers of Fab or H435A

variant (does not bind detectably to FcRn). Control IgG1 is an antibody directed against an irrelevant and non-disclosed target. Detection (AF650-labeled anti-FcRn Fab):inhibitor molar ratio at highest inhibitor concentration = 1:250 (optimized to measure competition with WT-IgG). Data represent histograms from a representative experiment, with indicated numbers showing gMFIs averaged from measured duplicates from the same experiment in (**b**), with aggregated data presented as mean values from three independent experiments in (**c**) and five in (**d**) using flow cytometry. Error bars indicate standard deviations. Statistical analysis was performed using a two-way ANOVA (Tukey's multiple-comparisons test) (**c**) and one-way ANOVA (Dunn's multiple-comparisons test) comparing to Fc-WT (**d**) and statistically significant differences are indicated by asterisks: *<0.05, **<0.01, ****<0.0001. ns not significant.

to PBS control ($P = 0.022$). Only Fc-MST-HN was significantly better in reducing tracer antibody levels compared with PBS ($P = 0.008$), while this was not significant for IgG1-MST-HN over time ($P = 0.255$). Fc-MST-HN showed both faster onset of the FcRn-antagonizing effect as well as a better overall elimination of the tracer antibody and was also found to be significantly different to the clearance induced by IgG1-MST-HN ($P = 0.039$) (Fig. 5c).

## Discussion

The binding of IgG to FcRn is an important process for the maintenance of IgG homeostasis. The biology is complex due to both the pH-dependent nature and steric considerations, which most likely include the IgG-Fab region[4,40–42,46], plasma membrane, and the context of different intracellular trafficking organelles.

Here we investigated the effect of the IgG-Fab region on IgG binding to membrane-associated FcRn. Using classical biochemical and noncellular methods, we found little or no effect of the Fab regions on FcRn binding. However, when probing the binding using cells, FcRn binding of IgG was negatively affected by the presence of the Fab arms, which translated into better FcRn occupancy and intracellular accumulation of the Fc fragments. In line with this, a Fc fragment with increased FcRn affinity (Fc-MST-HN) was better at blocking FcRn-

mediated IgG recycling compared to its full-size IgG counterpart, both in vitro and in non-human primates. These results suggest that the correct cellular membrane context of FcRn is crucial to investigate its true biological properties and suggest targeting FcRn with Fc fragments can provide stronger pharmacodynamic (PD) effects.

Subtle changes in affinity to human FcRn at acidic pH poorly predicts in vivo half-life of human IgG derivatives[43,53–55]. Wang et al., on the other hand, reported a correlation between the dissociation parameter $k_2/B$ determined at pH 7.3 in SPR and terminal in vivo half-life in humans[43]. Several groups have reported the IgG-Fab regions to impact the elution pH[41,49] in vitro, which was explained by the isoelectric point of the Fab arms[40]. These differ between antibodies mostly due to the differences in the charge distribution in the CDRs[41,42], which correlated with the in vivo half-life of the antibodies in human FcRn transgenic mice[40–43,49]. Here, we investigated the impact of the Fab arms in the context of antagonizing FcRn with variants with elevated FcRn affinity (MST-HN) by comparing an IgG1 variant with an IgG1-Fc only. We found no differences comparing IgG1 with IgG1-Fc molecules using SPR at either pH 6.0 or pH 7.4. This was the case for Fc-only and full-size IgG1 without mutations (WT), or FcRn enhancing mutations (MST-HN). Likewise, the Fab arms also had no impact on the elution pH as measured in FcRn-affinity chromatography, in which

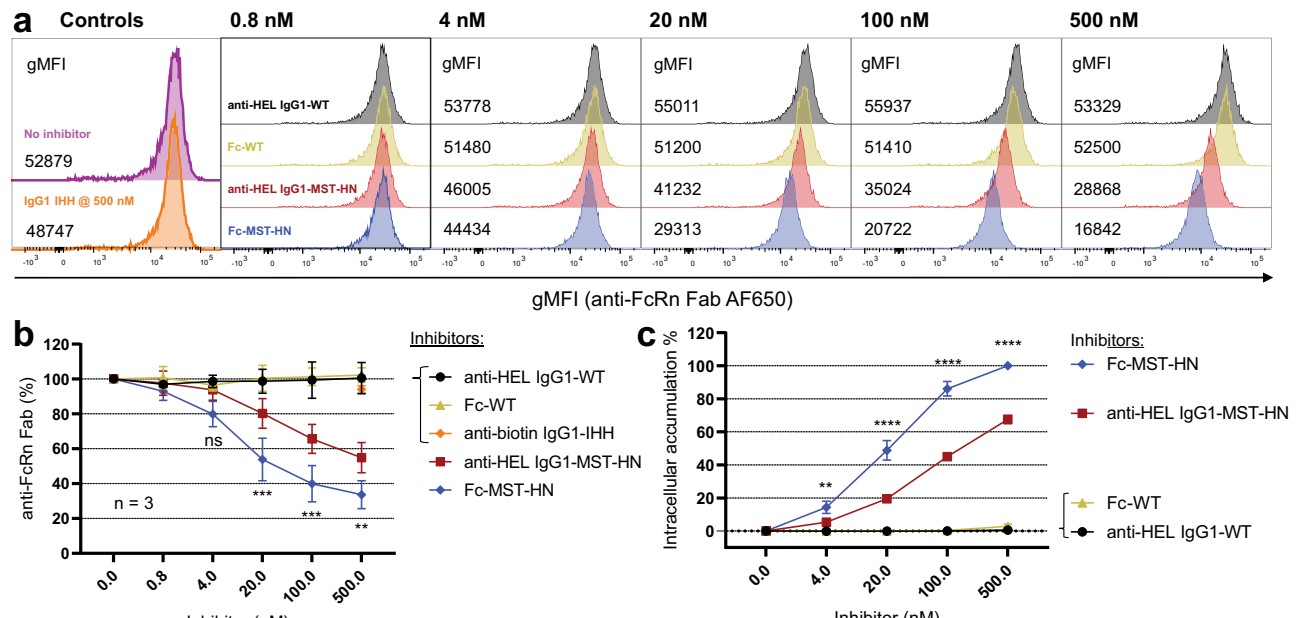

**Fig. 4 | Fc-MST-HN shows higher levels of intracellular FcRn occupancy compared with IgG1-MST-HN. a** Histograms of a representative experiment comparing the binding of AF650-labeled anti-FcRn Fab to FcRn in HEK-FcRn-GFP cells after pre-loading cells with indicated IgG variants at 37 °C for 2 h and subsequently removing surface-bound IgG by extensive washing. **b** Anti-FcRn Fab-AF650 staining of cells plotted as percent gMFI in the presence of indicated inhibitors compared with untreated cells. **c** Intracellular accumulation of equally AF405-labeled molecules in HEK-FcRn-GFP cells after pre-loading and washing the cells as in (**a**) as measured using flow cytometry. Data represent histograms from a representative experiment, with indicated numbers showing gMFIs averaged from measured duplicates from the same experiment in (**a**), with aggregated data presented as mean values from three independent experiments in (**b**) and (**c**). Error bars indicate standard deviations. Statistical analysis was performed using a two-way ANOVA (Sidak's multiple-comparisons test), and statistically significant differences are indicated by asterisks **<0.01, ***<0.001, ****<0.0001. ns not significant.

FcRn is coupled to the column material. Therefore, it seems that regardless of the setup's geometry, no effect of the Fab region are detected on FcRn binding when immobilizing one of the interaction partners. This suggests that the effect of the Fab arms on FcRn binding is observed when the spatial confinement of the cellular membrane is considered. Indeed, our crystallographic model showing the architecture of the Fc-MST-HN in complex with FcRn, confirms that although Fc molecules can bind FcRn in a 2:1 stoichiometry, a similar architecture involving whole antibodies would result in a steric clash between the membrane and the Fab arms. Confirmed by the lack of difference in the enthalpic component to explain the difference in FcRn affinity between Fc-MST and Fc-MST-HN[28,34,51], this model furthermore details how the F434 residue in the IgG-MST-HN increases the affinity in a pH-independent fashion by increasing the size of the hydrophobic-binding platform.

In line with the hypothesis that Fab arms only affect FcRn binding in the context of a cellular membrane, we found that when binding to membrane-associated FcRn at pH 6.0 on the cell surface, a negative influence of the anti-HEL Fab arms could be observed when comparing to IgG1-Fc only, and this for both WT and MST-HN variants. This was further corroborated for control antibodies of different specificities. Additionally, a stronger binding to membrane-associated FcRn was observed for IgG1 with less Fab arms (zero or one compared to two).

Using a slight modification of a recently described FcRn occupancy assay[32], we found that not only binding of IgG1-MST-HN to FcRn on the cell surface is affected by the Fab arms, but also the intracellular accumulation and subsequently intracellular FcRn occupancy as found in the recycling tubules and vesicular compartments[22–24]. This implies that this potential steric hindrance might play an even bigger role intracellularly due to the possible curvature of the membrane in recycling vesicles, but also in recycling tubules where the Fab arms are likely to be sterically interfering with efficient binding with FcRn in a lying down orientation[24,52,56,57]. No signs of non-specific binding of the Fc only were observed compared with full-size IgG to neither of the column materials (Fig. 1c, Supplementary Figs. 1a and 6). This taken together with the FcRn-specific nature of our competition-based cellular assays and the use of the FcRn-dead controls (Figs. 3 and 4) support the conclusion that the effect we observe is truly FcRn-dependent. In summary, the data suggest that IgG binding to FcRn in the context of the cellular membrane is impeded by the physical restrictions imposed by the two Fab regions (Fig. 6a). This is evident when modeled using actual crystal structures (Fig. 6b1), which illustrates that IgG binding requires bending of the IgG and/or FcRn (Fig. 6b2), as suggested elsewhere[4,45,46]. This is also in agreement with the possibility that FcRn-IgG interactions occur on parallel membranes in a 2 FcRn:1 IgG fashion, possibly in intracellular tubular FcRn-routing organelles[52,57]. Although both, 1:1 and 2:1 FcRn:IgG stoichiometries have been suggested[58–61], several groups have found that for full binding capacity and in vivo activity, both Fc halves are required[19,47,57,62,63], which is why we assume a 2:1 IgG:FcRn stoichiometry.

The IgG/Fc uptake in our HEK-FcRn-GFP cell line, which also expresses FcRn on the cell surface, was FcRn-dependent, making it formally impossible to directly distinguish the negative contribution of the Fab arms on surface binding and uptake/intracellular accumulation from the impact on binding to intracellular FcRn.

Previous efforts investigated FcRn-dependent processes making use of FcRn-overexpressing cell lines in vitro[22,23,34,45,53,64]. These include the A375-FcRn cells[45,64], MDCK-FcRn cells[53] as well as HEK-FcRn-GFP cells used in this study, which all show surface expression of FcRn. Grevys et al. determined whole FcRn expression after transfection, but they reported higher uptake of IgG at pH 6.0 when compared to loading at pH 7.4[34], also pointing towards FcRn surface expression. In the same FcRn-overexpressing HMEC-1 system, Gjølberg et al.[65] observed FcRn-dependent uptake of the WT-Fc only, but not of WT IgG. This aligns with our data presented here, as results of both studies

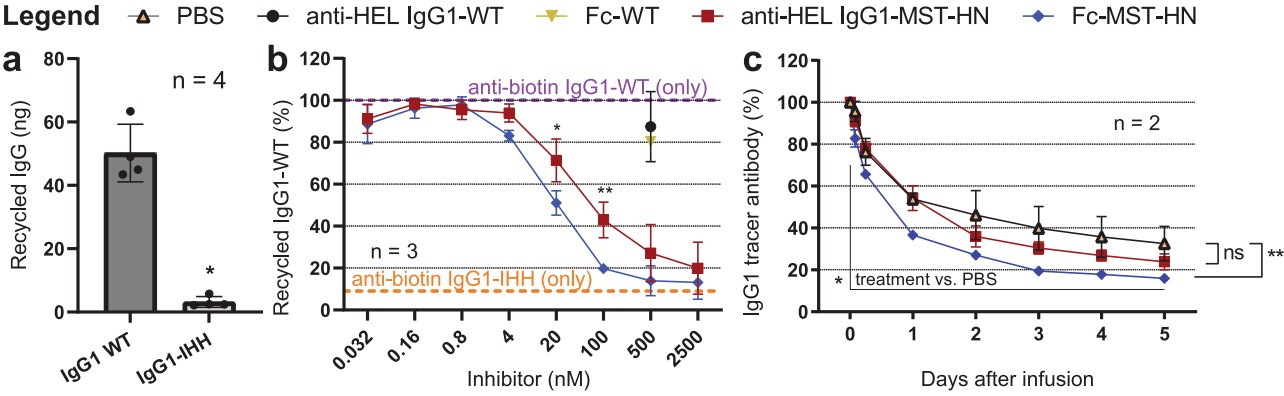

**Fig. 5 | Fc-MST-HN inhibits FcRn-mediated recycling of IgG in vitro and reduces IgG tracer levels in cynomolgus monkeys more efficiently than IgG1-MST-HN.** **a** Absolute amounts of recycled anti-biotin IgG1-WT and anti-biotin IgG1-IHH in experiments using HEK-FcRn-GFP cells as measured by ELISA. **b** Relative amount of recycled anti-biotin IgG1-WT. Cells were preloaded with the inhibitors at different concentrations, washed extensively, and then loaded with a fixed concentration of anti-biotin IgG1-WT, again washed extensively, and incubated under culture conditions. Recycled amounts of anti-biotin IgG1-WT were quantified by ELISA the next day. Data represent aggregated data presented as mean values from four independent experiments in (**a**) and three in (**b**). Data in (**b**) are normalized to medium control (in two experiments) or point of least inhibition. Error bars indicate standard deviations. **c** Cynomolgus monkeys were assigned to groups of two animals per group and injected with 1 mg/kg anti-mouse CD70-hIgG1 tracer antibody 5 min prior to infusion of PBS or equimolar amounts per kg of Fc-MST-HN or anti-HEL IgG1-MST-HN. Tracer antibody levels were plotted as percentage relative to pre-dose (mean ± SEM). Statistical analysis in (**a**) was performed using a two-tailed Mann–Whitney test and a two-way ANOVA (Sidak's multiple-comparisons test) in (**b**). Statistical analysis in (**c**) was performed using a two-sided Walt test on the logarithm of the relative tracer antibody levels using a longitudinal model with treatment group, time post-infusion and their interaction as fixed effects, and a heterogeneous first-order autoregressive covariance structure. The model was run with restricted maximum likelihood and the Kenward–Roger approximation; for the denominator, degrees of freedom were applied. Closed testing procedure was applied to control the type 1 error rate at 5%, no multiple testing correction was needed. Statistically significant differences are indicated by asterisks *<0.05, **<0.01. ns not significant.

indicate that FcRn-dependent uptake is affected by the Fab. However, this surface expression of FcRn in these systems does not limit the relevance of these assays, as they have proven to reflect FcRn-dependent processes in vivo[34,53,64].

We employed our HEK-FcRn-GFP cells to compare the blocking efficacies of MST-HN variants in an FcRn-dependent IgG recycling assay. Those results also indicated better uptake and blocking efficacy using an Fc only, suggesting that Fab fragments impede uptake, intracellular accumulation and blocking capacity. More importantly, those results were confirmed in cynomolgus monkeys, where hIgG1-WT tracer antibody was used to model pathogenic IgG. Using the MST-HN-IgG1 variants, we found that these indeed facilitated the targeted lysosomal degradation of the tracer antibody, as intended by their use as FcRn-antagonizing compounds to treat IgG-mediated autoimmune diseases[27,29]. Clearance of the tracer antibody in the groups treated with Fc-MST-HN and IgG1-MST-HN exceeded tracer clearance in the PBS control group, confirming the principle of the approach, as described previously[29,31,32,35]. Human IgG has been reported to be cleared from circulation about two times faster in cynomolgus monkeys than in humans, which explains the relatively rapid clearance of the tracer antibody in the control animals[54]. Infusion of Fc-MST-HN led to a significantly faster clearance of the tracer antibody when compared to anti-HEL-IgG1-MST-HN as well as a deeper maximum drop in tracer hIgG1 levels. The tendency of a faster onset of the FcRn-antagonizing effect is in line with higher intracellular accumulation of the Fc-MST-HN in vitro in these cells with FcRn expressed on the surface, and might suggest more efficient receptor-mediated internalization in cells is responsible for the recycling of IgG[7]. Gjølberg et al.[65] also found that Fc-WT has a shorter apparent serum half-life than its full-size IgG counterpart in hFcRn transgenic mice. Our results indicate more efficient FcRn blocking by the Fc-MST-HN than anti-HEL IgG1-MST-HN. The two studies are complementary, both showing that an easier FcRn-dependent cellular entry of Fc only, allowing better access of FcRn within cells. For both WT-Fc and Fc-MST-HN this results in enhanced intracellular retention compared to full-sized IgG.

IgG internalization has been suggested in the past to occur through pinocytosis[66,67]. Clearly, the overall FcRn-mediated IgG recycling pathway is complex, which includes processes ranging from short to prolonged release. During these processes, FcRn containing exocytotic vesicles have been observed to completely fuse with the membrane or repeatedly partially fuse and thereby release their IgG cargo in a stepwise fashion[22,68]. We speculate that during such events, in which intravesicular pH might remain acidic, a full-size IgG is more likely to be released from the vesicles than a Fc-only, presumably due to steric impedance resulting from the Fab regions, as evident from our findings. Accordingly, we hypothesize that full-size IgGs have a longer apparent serum half-life compared to Fc only, as evident from ref. [65] and previous findings[63]. However, in the context of antagonizing FcRn with an engineered Fc fragment, having Fabs as a steric entity would lead to less efficient blocking of FcRn. Furthermore, it seems that Fc-MST-HN remains bound to FcRn rather than actually being cleared. This would also explain the relatively short apparent serum half-life of Fc-MST-HN in vivo in cynomolgus monkeys and humans while retaining strong biological activity, possibly because of prolonged intracellular retention[29].

In this study, we confirm the relevance of FcRn-dependent in vitro assays to predict in vivo behavior of IgG molecules[34,53,69]. We describe a fast and easy assay to measure binding of IgG1 molecules to membrane-associated FcRn, and we show that the results correlate with intracellular accumulation in vitro and data obtained from cynomolgus monkey experiments. We conclude that the cellular context, most likely the association with the cell membrane, plays a crucial role in FcRn biology. Its interplay with the Fab arms, or other cargo as utilized in Fc-fusion-based drugs is an important yet largely overlooked factor, which needs to be considered for the development of therapeutic antibodies and other IgG-Fc-based fusion proteins.

## Methods

### Generation of HEK-FcRn-GFP cells

HEK cells stably expressing human FcRn-GFP fusing protein (HEK-FcRn-GFP cells) were generated from HEK293 cells (CLS Cat. No. 3000192). DNA strands encoding human FcRn alpha chain with a GFP tag and human ß2-microglobulin were ordered from Geneart (Thermo Scientific). Following a two-step cloning protocol, both constructs

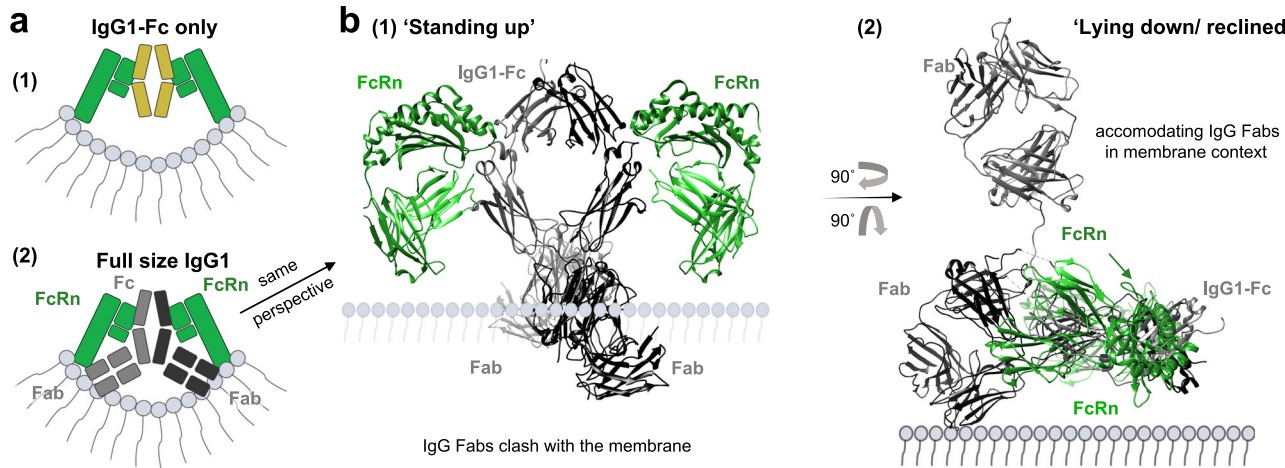

**Fig. 6 | IgG-Fab arms cause a steric hindrance when IgG is bound to membrane-associated FcRn. a** Graphical illustration of potential steric hindrance caused by the Fab arms when an IgG binds membrane-associated FcRn. **b** Crystal structure of FcRn bound to IgG in a membrane context in the same "standing up" conformation as in (**a2**) and in the "lying down/reclined" conformation. The images were rendered using UCSF Chimera[97] after combining structural data from ref. [98] (PDB: 1HZH) and ref. [25] (PDB: 1I1A) by comparative homology modeling using the matchmaker function of UCSF Chimera.

were cloned into a pVITRO1-mcs expression vector (InvivoGen) containing a Geneticin resistance gene. In total, $4 \times 10^5$ HEK293 cells were seeded in a six-well plate (Corning), followed by transfection of the cells 24 h after seeding using a mix of 2 μg DNA, 150 μL Optimem (Life Technologies) and 6 μL fuGENE (Promega). Thirty minutes after transfection, 0.5 mL of DMEM (Sigma) supplemented with 10% FCS (Sigma) was added, and cells were cultured for 24 h at 37 °C, 5% $CO_2$. Cells were subsequently washed and transferred to 10-cm² dishes (Thermo Scientific), adding 500 μg/mL Geneticin (Life Technologies) as selection antibiotic. Cells were then cultured for 7 days, exchanging medium every two days. GFP expression of cells was confirmed using flow cytometry. Following another 14 days in culture, exchanging the medium every 2 days, colonies were subsequently transferred to a 24-well plate (Corning) using 8 × 8 mm cloning cylinders (Sigma) according to the manufacturer's protocol. Following another 14 days in culture exchanging the medium every 2 days, FcRn expression was confirmed by both GFP signal and APC-labeled Nb218-H4[70], using flow cytometry. Positive clones were expanded and frozen according to standard procedures.

### Cells
HEK293F cells (Thermo Fisher Scientific) were cultured in Erlenmeyer flasks (Corning) in 293 Freestyle expression medium (Thermo Fisher Scientific) at 37 °C, 8% $CO_2$, and shaking at 125 RPM.

HEK-FcRn-GFP cells were cultured in RPMI1640 (Thermo Scientific) supplemented with 10% (v/v) FCS, 4 mM L-Glutamine (Thermo Scientific), 100 U/mL penicillin, and 100 μg/mL streptomycin (Thermo Scientific), which will be referred to as "fully supplemented culture medium", in Nunclon Delta Surface 80-cm² flasks (Thermo Scientific) and passaged with 0.05% (m/v) Trypsin with 0.02% (m/v) EDTA (Thermo Scientific) when 80–90% confluence was reached.

### Cloning of antibodies and recombinant human FcRn
Linear DNA strands encoding for IgG1-WT, MST-HN, and IHH (I253A, H310A, H435A: a triple AA substitution in IgG1 known to abrogate FcRn binding[48,71]) of the IGHG1*03 allotype were ordered from Integrated DNA Technologies and cloned into a pcDNA3.1 expression vector containing anti-biotin heavy chain variable regions obtained from[72,73], as described previously[74]. Linear DNA strands encoding for IgG1 of the IGHG1*03 allotype containing the complementary Duobody® mutations K409R and F405L[75] were ordered and processed similarly. These were ordered as Fc-only, anti-irrelevant specificity (Control IgG) and the same control IgG with an additional H435A mutation to ablate FcRn

binding[76]. Linear DNA strands encoding for the FcRn α-chain with a C-terminal BirA 10x-Histidine (His) tag and ß2-microglobulin were ordered accordingly and cloned into a pcDNA3.1 expression vector, as described elsewhere[49]. In brief, expression vectors and DNA inserts were digested with EcoRI and NheI FastDigest restriction enzymes (Thermo Scientific). The expression vector backbone was isolated by gel purification using a 1% UltraPure agarose (Thermo Scientific) gel with 1:10,000 SYBR Safe (Invitrogen). DNA was extracted from the gel using the NucleoSpin Gel and PCR clean-up kit (Macherey-Nagel) according to the manufacturer's protocol. The DNA fragments were isolated using the same kit but without prior gel purification.

The DNA fragments were ligated into the pcDNA3.1 backbone overnight at 16 °C using T4 DNA Ligase (New England Biolabs) in 1x T4 DNA Ligation buffer (New England Biolabs) with a 3:1 insert to vector molar ratio. In total, 5 μL of the ligation reaction was transformed into 50 μL DH5α competent cells (Thermo Scientific) by heat shock.

The cells were plated on LB-agar plates containing 50 μg/mL ampicillin (Thermo Scientific) and incubated overnight at 37 °C. Colonies were picked and grown in 2 mL LB medium containing 50 μg/mL ampicillin (Thermo Scientific) overnight at 37 °C, shaking at 180 RPM. DNA was isolated from the bacterial culture using the NucleoSpin Plasmid EasyPure kit (Macherey-Nagel) according to the manufacturer's protocol and sequenced.

Following confirmation of sequences, DNA constructs were used for the transformation of DH5α competent cells as described above, and colonies were picked and grown in a 5 mL pre-culture. This was subsequently used to inoculate 200 mL LB medium containing 50 μg/mL ampicillin (Thermo Scientific), after which the culture was grown overnight at 37 °C, shaking at 180 RPM. DNA was isolated using the NucleoBond Xtra Maxi kit (Macherey-Nagel) according to the manufacturer's protocol and again sequenced.

### Production of in-house recombinant proteins
Antibodies were produced as described previously[74]. In brief, 31.35 μg of the heavy chain vector, 37.65 μg light chain vector (pcDNA3.1 anti-biotin VLCL[72,73], and 31 μg pSVLT/p21/p27 mix[77] were added to 6.66 mL opti-MEM (Thermo Scientific) per 100 mL of transfection cell culture. 300 μL Polyethylenimine (PEI) MAX (linear, MW 4.000, Polysciences) was added, the mixture was immediately vortexed and incubated for 20 min at room temperature. 100 mL of HEK293F cells (Thermo Scientific) at $1 \times 10^6$ cells/mL in fresh FreeStyle 293 Expression Medium (Thermo Fisher Scientific) were transfected with the mixture and incubated at 37 °C at 8% $CO_2$ and shaking. After 4 h, 100 units/mL

penicillin and 100 μg/mL streptomycin (Thermo Fisher Scientific) were added to the culture. Human FcRn was produced as described previously[49], using equimolar amounts of both expression vectors. The culture supernatants were harvested 6 days after transfection by spinning down the cells twice for 5 min at 3100 × *g* and filtering through a 0.45-μm syringe filter (Whatman).

For crystallization and ITC purposes, the cDNA sequence of FcRn residues 1–297 (UniProt entry P55899) was cloned in the pHLsec vector[78] to allow transient expression of the extracellular domain with a C-terminal thrombin-cleavable AviTag, followed by a hexahistidine sequence. The cDNA sequence of beta-2-microglobulin (ß2M) residues 1–119 (UniProt entry P61769), followed by a stop codon, was cloned in the pHLsec vector to allow tagless expression. HEK293S MGAT1$^{-/-}$ TR$^+$ cells[79] were grown to 90% confluence in high glucose DMEM medium supplemented with 10% FCS. Before transfection, medium was exchanged for serum-free medium. 25 kDa branched PEI was used as transfection reagent[78]. Equal amounts of expression plasmids encoding β2M and FcRn were used for co-transfection.

### Purification of recombinant antibodies

Antibodies were purified from culture medium with AKTA prime (GE Healthcare) by affinity chromatography using either a 5 mL HiTrap HP protein A (IgG1-WT and -MST-HN) or protein G (IgG1-IHH) column (GE Healthcare), as described previously[80], or on a HisTrap HP column (FcRn) (GE Healthcare). Fractions containing the antibodies or FcRn were combined and concentrated using a 10 K MWCO Pierce Protein Concentrator PES (Thermo Scientific). Antibodies were fractionated by HPLC-SEC using an AKTA UPC-900, P-920 and Frac-950 (GE Healthcare) with a Superdex 200 10/300 GL column (GE Healthcare). Monomeric fractions were combined, and antibodies were dialyzed against 5 mM sodium acetate (pH 4.5) and human FcRn against 1 × PBS using either a 10 K MWCO Slide-A-Lyzer dialysis cassette (Thermo Scientific) overnight at 4 °C or using a 7 K MWCO Zeba Spin desalting column (Thermo Scientific) according to the manufacturer's protocol. Protein concentrations were measured using a Nanodrop 2000c spectrophotometer (Thermo Scientific), adjusted to 1 mg/mL and aliquoted to 20 μL working stocks. Working stocks were stored at −20 °C until assayed. Production and purification of anti-HEL IgG1-WT, anti-HEL IgG1-MST-HN, Fc-WT, Fc-MST-HN, anti-IgE IgG1-MST-HN[81] and anti-FcRn-IgG blocking Fab (SYNT001)[82] (anti-FcRn Fab) were outsourced for production by Evitria SA (Switzerland), as described previously[29].

### Purification of recombinant FcRn in complex with ß2M

Following 5 days of transient expression, the conditioned medium was harvested, cleared of cellular debris by centrifugation and filtered through a 22 μm cut-off bottle top filter. Recombinant hexahistidine-tagged FcRn was captured from the conditioned medium by IMAC purification using a cOmplete His-Tag purification column (Roche). After elution with 500 mM imidazole, the eluate was concentrated and further purified by size-exclusion chromatography using HiLoad 16/60 Superdex 75/200 columns (GE Healthcare) with Bis−Tris buffer (50 mM Bis-TRIS pH 6.0; 150 mM NaCl) as running buffer. Protein purity was evaluated by SDS-PAGE. Size-exclusion chromatography coupled online to a multi-angle laser light scattering device (SEC-MALLS) and/or biolayer interferometry (BLI) were used to qualitatively validate binding of the produced FcRn:ß2M to Fc-WT, Fc-MST, and Fc-MST-HN.

### SDS-PAGE and HPLC-SEC

Antibodies, antibody fragments, and recombinant human FcRn were analyzed using SDS-PAGE run under reducing and non-reducing conditions, as described previously[74]. Additional analytical HPLC-SEC runs were performed using an Agilent 1260 Infinity II HPLC system (Agilent) coupled to a SDP-20A UV/Vis detector (SHIMADZU), a miniDAWN

(Wyatt Technologies), and an Optilab (Wyatt Technologies). 16.67 μg of each molecule were assayed using a Superdex 200 10/300 GL (GE Healthcare) in 1xPBS (Fresenius Kabi) at a flow rate of 0.75 mL/min.

### SPR

Affinity measurements for binding to human FcRn using surface plasmon resonance (SPR) were performed using an IBIS MX96 (IBIS Technologies) device and a Continous Flow Microspotter (Wasatch Microfluidics). Antibodies and their fragments were spotted at four concentrations each on a SensEye G Easy2Spot (SensEye) in 10 mM Sodium acetate at pH 4.5 with 0.075% (v/v) Tween (80) in 2× dilution series starting at 60 nM or 180 nM, respectively. Kinetic titration of human FcRn was performed by injecting twelve concentrations of human FcRn, from 0.49 nM to 1000 nM (2x dilution series) in 1 × PBS containing 0.075% (v/v) Tween (80) at pH 7.4 or pH 6.0. The sensor was regenerated between the cycles using two subsequent injections of 20 mM Tris-HCl, 150 mM NaCl pH 8.8 and 20 mM H$_3$PO$_4$ pH 2.4. $K_D$ values were calculated by performing an equilibrium analysis and fitting a Langmuir 1:1 binding model to a R$_{max}$ = 700 RU, as described previously[83], using Scrubber software version 2 (BioLogic Software) and Excel. 1:1 binding model was used as IgG-based molecules were coupled to the chip, assuming each FcRn molecule has only one binding site for IgG.

### FcRn-affinity chromatography

FcRn-affinity chromatography was performed according to the manufacturer's instructions using a FcRn Affinity Column (Roche) on an ÄKTA pure M1 (GE Healthcare). In brief, 100 μg of antibody or antibody fragment was diluted in low pH MES buffer (20 mM MES (Sigma), 150 mM NaCl (Sigma), pH 5.5) and injected into the column. A pH gradient was applied over a total volume of 50 mL by injecting high pH elution buffer (20 mM Tris-HCl (Thermo Scientific), 150 mM NaCl (Sigma), pH 8.8) to determine retention times and elution pHs of the variants, as published previously[34,49]. Additional FcRn-affinity chromatography experiments were outsourced to RIC biologics (Belgium) and performed in the same manner, but using acetate buffers at the same pH as described above and injecting a total amount of 12 μg per molecule. UV 280 nM signals were normalized to the maximum response of each molecule and plotted over the retention time.

### Crystal structure determination of Fc-MST-HN and its complex with FcRn:β2M

For the crystallization of Fc-MST-HN in complex with FcRn, recombinant FcRn:β2M was treated with EndoH (New England Biolabs) to trim N-linked glycosylation[84] and with 1 U/μg protein thrombin (New England Biolabs) to remove purification tags. EndoH and thrombin were removed by preparative SEC. Protein purity was evaluated by SDS-PAGE. Complexes between FcRn:β2M and Fc fragments were biochemically reconstituted by adding the Fc fragment to substoichiometric amounts to FcRn:β2M, followed by isolation of the complex using preparative SEC. Protein purity was evaluated by SDS-PAGE. Optimal concentrations for protein crystal generation and growth of complexes between FcRn:β2M and Fc-MST-HN were found to be 3.2–3.3 mg/ml, and 8 mg/ml for Fc-MST-HN alone. For both the complex and the Fc alone, vapor-diffusion crystallization experiments were set up using a Mosquito crystallization robot (SPT Labtech) in nano-liter scale SwissSci 96-well triple-drop plates. Protein plates were incubated at 293 K. Commercially available sitting drop crystallization screens (Molecular Dimensions, Hampton Research) were used to screen for candidate crystallization conditions. Optimized crystals of FcRn:β2M:Fc-MST-HN were grown from 0.1 M sodium cacodylate pH 7.0 and cryo-cooled by direct plunging into liquid nitrogen without additional cryoprotectant. Optimized crystals of Fc-MST-HN were grown from 0.2 M CaCl$_2$, 20% (w/v) PEG 3350 at pH 5.1, cryoprotected

with mother liquor supplemented with 20% ethylene glycol, and cryo-cooled by direct plunging into liquid nitrogen. For both datasets, diffraction data were indexed, integrated and scaled using the XDS suite[85]. When indicated, the resulting datasets were truncated and rescaled using the STARANISO anisotropy & Bayesian estimation server[86]. Initial phases were obtained using maximum-likelihood molecular replacement by Phaser from the CCP4 package[87,88]. Search models were generated from re-refined X-ray structures of Fc-MST in complex with FcRn (PDB ID 4N0U), truncated to the last common Cα[50]. Structure building and refinement was performed iteratively using COOT[89], PHENIX[90] and BUSTER[91]. Crystallographic coordinates and data have been deposited in the PDB (www.rcsb.org) with accession codes 7Q15 and 7Q3P.

### Binding thermodynamics for the interaction of Fc-MST and Fc-MST-HN with FcRn

Aliquots of human Fc-MST, Fc-MST-HN and FcRn were thawed prior to use and buffer exchanged using a HiTrap 5-ml desalting column (GE Healthcare). Protein concentrations were estimated spectrophotometrically using their extinction coefficients. All ITCs were performed at 37 °C using a PEAQ-ITC (Malvern Panalytical). When titrating the Fc in FcRn, concentrations were determined to be 9 µM and 52.7 µM for the cell and syringe, respectively. In the reverse setup, concentrations of 6 µM and 165 µM were used for respectively the cell and syringe. In each experiment, a total of 18 2 µl-titrations with varying spacing time were preceded by an initial injection of 0.4–0.8 µl. Data were analyzed using NITPIC version 1.3.0[92,93] using the default parameters. Calculated heats and error estimates of all injections were spawned to Sedphat version 1.2[94]. Interactions were modeled using the "$A + B + B < - > \{AB\} + B < - > ABB$ with 2 symmetric sites, macroscop K" model with the following global parameters: incf$A$ = incf$B$ = 0, not refined; Log($Ka1$) = 6, refined; dH$AB$ = −10, refined; Log10($Ka2/Ka1$) macroscopic = −0.6, not refined; dH($AB$)$B$-dH$AB$ = 0, not refined. Under "experiment parameters", it was allowed to fit a baseline and estimate a local correction factor for the cell concentration. Buffer, pH and temperature were completed and, given our setup, we selected "titrate $A$ into $B$". After a global fit, the estimated thermodynamic parameters of the $ABB$ reaction were calculated from those of the $AB$ estimates. Finally, GUSSI version 1.4.2 was used to generate a figure containing the thermogram and isotherms[95]. For the generation of the final figures, representative ITC thermograms were selected and overlaid with the calculated thermodynamic parameters of the interactions.

### FcRn surface competition assay

HEK-FcRn-GFP cells were trypsinized as described above and the reaction was stopped with fully supplemented culture medium. The assay was performed on ice and/ or at 4 °C. Cells were subsequently washed twice with ice-cold 1 × PBS containing 1% (v/v) FCS at pH 6.0 and adjusted to 2 × 10^6 cells/ mL. Fivefold dilution series of fourfold concentrated antibodies/antibody fragments to be tested were prepared in ice-cold 1 × PBS containing 1% (v/v) FCS at pH 6.0. A stock solution of 80 nM Dylight 650-labeled (Thermo Scientific, Cat. No. 62265) anti-FcRn Fab[82] was prepared in the same buffer. In total, 25 µL of antibody or antibody fragment and anti-FcRn Fab solutions were combined in a sterile 96-well U-bottom plate (Corning) and mixed well. In all, 50 µL of the above-mentioned HEK-FcRn-GFP cell suspension was added, reaching final concentrations of 20 nM for the Dylight 650-labeled anti-FcRn Fab, 500 nM at the highest concentration of the fivefold dilution series of the examined molecules and 10^5 cells per well. Subsequent experiments with control IgG1 Duobody® containing two Fab arms and the H435A substitution, two Fab arms, one Fab arm, anti-HEL IgG1-WT and Fc-WT were performed in the same manner, using a final concentration of 2 nM of the Dylight 650-labeled anti-FcRn Fab and 500 nM of all IgG1-Fc containing

molecules. The plate was incubated on ice for 30 min in the dark and subsequently spun at $450 \times g$ at 4 °C. Cells were washed once with 150 µL ice-cold 1 × PBS containing 1% (v/v) FCS at pH 6.0, spun down as described above, and washed with 150 µL ice cold 1 × PBS containing 1% (v/v) FCS at pH 7.4. Cells were fixed using the BD Cytofix/Cytoperm fixation/permeabilization kit (BD Biosciences) according to the manufacturer's instructions. Cells were then washed twice with 150 µL ice-cold 1 × PBS containing 1% (v/v) FCS at pH 7.4, suspended in 100 µL of the same buffer, and measured using flow cytometry. FcRn surface expression on HEK-FcRn-GFP cells and FcRn specificity of the anti-FcRn Fab was verified by surface staining as described above in comparison to a Dylight 650-labeled (Thermo Scientific) human Fab isotype (Novus Biologicals) with a comparable degree of labeling (~10% difference).

### Intracellular FcRn occupancy assay

The protocol was adapted from ref. 32. In all, 10^5 HEK-FcRn-GFP cells per well were loaded with a fivefold dilution series of antibodies to be tested in RPMI1640 (Thermo Scientific) supplemented with 10% (v/v) FCS, 4 mM L-glutamine (Thermo Scientific), 100 U/mL penicillin and 100 µg/mL streptomycin (Thermo Scientific) and incubated at 37 °C and 5% $CO_2$ for 2 h in a sterile 96-well U-bottom plate (Corning). The plate was then put on ice for 2 min and subsequently spun at 4 °C at $450 \times g$ for 3 min and the supernatants were discarded. Cells were transferred to a 96-well V-bottom plate (Corning) and washed five times with 150 µL ice-cold medium, followed by centrifugation at 4 °C at $450 \times g$ for 3 min. Cells were then washed once with 150 µL 1 x PBS containing 1% (v/v) FCS at pH 6.0, followed by centrifugation at 4 °C at $450 \times g$ for 3 min. At the day of the assay, both solutions of the BD Cytofix/Cytoperm Fixation/Permeabilization Kit (BD Biosciences) were adjusted to pH 6.0 after the addition of MES (Thermo Scientific) to a final concentration of 0.5 M. Cells were fixed and performed according to the manufacturer's protocol. Staining for available FcRn was performed using 1 µg/mL of Dylight 650 (Thermo Scientific) labeled anti-FcRn Fab[82] in Cytoperm solution (BD Biosciences) at pH 6.0 for 30 min on ice in the dark. Cells were washed twice with 150 µL 1 x PBS containing 1% (v/v) FCS at pH 6.0, followed by centrifugation at 4 °C at $450 \times g$ for 3 min.

### Uptake/intracellular accumulation experiments

Anti-HEL IgG1-WT, anti-HEL IgG1-MST-HN, Fc-WT, and Fc-MST-HN were labeled with Dylight 405 (Thermo Scientific, Cat. No. 46400) according to the manufacturer's protocol. Labeling reactions were titrated in order to achieve comparable degrees of labeling. Overall, 10^5 HEK-FcRn-GFP cells per well were loaded with a 5× dilution series starting at a concentration of 500 nM equally labeled antibodies to be tested in fully supplemented RPMI1640 (Thermo Scientific) and incubated at 37 °C and 5% $CO_2$ for 2 h in a sterile 96-well U-bottom plate (Corning). The plate was then put on ice for 2 min and subsequently spun at 4 °C at $450 \times g$ for 3 min, and the supernatants were discarded. Cells were transferred to a 96-well V-bottom plate (Corning) and washed five times with 150 µL ice-cold medium, followed by centrifugation at 4 °C at $450 \times g$ for 3 min. Cells were then washed twice with 150 µL 1xPBS containing 1% (v/v) FCS at pH 7.4, followed by centrifugation at 4 °C at $450 \times g$ for 3 min. Cells were resuspended in 100 µL 1xPBS containing 1% (v/v) FCS at pH 7.4 and analyzed using flow cytometry.

### Flow cytometry

Flow cytometry measurements were performed using BD FACSCanto-II Cell Analyzer (BD Biosciences), BD LSR-II (BD Biosciences) or BD LSRFortessa (BD Biosciences) using BD FACSDiva Software (BD Biosciences). 10.000 events in the gate of interest were acquired per sample. Data were analyzed using FlowJo 10.7.1 Software (BD Biosciences).

## Recycling assay

The recycling assay was inspired by and adapted from[34]. A sterile 96-well flat bottom plate (Corning) was coated with 50 µL 0.01% (m/v) 0.22 µm filtered (Whatman) Poly-L-Lysine (MW 70,000–150,000) (Sigma) solution per well for 1 h at room temperature. The solution was removed and the plate dried under sterile conditions for 1 h at RT. Subsequently, $1.2 \times 10^5$ HEK-FcRn-GFP cells were seeded per well and cultured at 37 °C at 5% $CO_2$ overnight. The next morning, confluency was confirmed. Medium was removed carefully, and cells were starved in 1 × HBSS (Gibco) at 37 °C at 5% $CO_2$ for 1 h. In the meantime, 5 × dilution series of anti-HEL IgG1-MST-HN, as well as Fc-MST-HN, were diluted in 1 × HBSS starting at 2500 nM to 0.032 nM. Furthermore, solutions of anti-HEL IgG1-WT and Fc-WT in 1 × HBSS (Gibco) were prepared at 500 nM. After starving the cells, the supernatants were removed and the antibody or antibody fragment titration series were added as well as 500 nM of the anti-HEL IgG1-WT and Fc-WT, next to 1xHBSS (Gibco) only. The cells were subsequently incubated at 37 °C at 5% $CO_2$ for 2 h. After this loading step, the plate was washed 5× with 150 µL ice-cold 1 × HBSS, followed by one washing step with 150 µL ice-cold 1 × HBSS containing 20 mM MES (Sigma) at pH 6.0. Anti-biotin IgG-WT and anti-biotin IgG1-IHH were prepared at 500 nM in 1 × HBSS (Gibco) containing 20 mM MES (Sigma) at pH 6.0, added to specific wells, and incubated at 37 °C and 5% $CO_2$ for 2 h. Afterward, cells were washed 5× with ice-cold 1 × HBSS (Gibco) at pH 7.4, 150 µL prewarmed RPMI1640 w/o supplements were added carefully, and the cells were incubated overnight at 37 °C at 5% $CO_2$. The next morning (16 h later), 125 µL of each supernatant was taken off and measured in ELISA.

## Anti-biotin ELISA

Ninety-six-well Nunc MaxiSorp plates (Thermo Scientific) were coated with 100 µL of 1 µg/mL biotin-BSA (3 biotin/ BSA) (ITK Diagnostics) in 1 × PBS (Fresenius Kabi) overnight at 4 °C. The plates were washed 5× with 1xPBS containing 0.05% (v/v) Tween using a BioTek 405 LS plate washer (BioTek Instruments). Plates were blocked with 2% (m/v) milk (Campina) in 1 × PBS containing 0.05% (v/v) Tween shaking at 350 RPM at RT for 1 h. Plates were again washed 5× with 1 × PBS containing 0.05% (v/v) Tween. In all, 3× dilution series of the supernatants of the recycling assay as well as standards of anti-biotin IgG1-WT and -IHH in 1 × PBS containing 0.05% (v/v) Tween were added to the plates and incubated for 1 h at RT shaking at 350 RPM. Plates were washed as previously, 100 µL of 1/1000 mouse anti-human IgG-Fc-HRP (Southern Biotech, Cat. No. 9040-05) in 1 × PBS containing 0.05% Tween was added and incubated for 1 h at RT shaking at 350 RPM. After washing, ELISAs were developed with 3,3'5,5'-tetramethylbenzidine (TMB) (Merck) and reactions were stopped with 100 µL 2 M $H_2SO_4$. ELISAs were read out in BioTek Synergy 2 (BioTek Instruments) at 450 nM.

## Cynomolgus monkey experiments

The cynomolgus monkey experimental protocols were approved by local authorities (Niedersächsisches Landesamt für Verbrauchschutz und Lebensmittelsicherheit, Wardenburg, Germany) and conducted according to local regulations. Captive-bred female cynomolgus monkeys (R.C. Hartelust BV) were acclimatized in the test facility (Provivo Biosciences, Hamburg, Germany) for four weeks prior to the study. After veterinary examination, animals were assigned to study groups of two animals per group. The animals were 3.1 to 3.5 years of age at the first dosing. Experiments were conducted as described previously[29]. Briefly, animals were intravenously injected with 1 mg/kg anti-mouse CD70-hIgG1 tracer antibody 5 min prior to an intravenous infusion of a single dose of 20 mg/kg anti-HEL IgG1-MST-HN or an equimolar amount of Fc-MST-HN or PBS. Blood was sampled from saphena magna of the animals at different time points and serum fractions were obtained by centrifugation. Tracer antibody levels were determined by murine CD70-binding ELISA-specific ELISA.

## Statistical analysis and data presentation

Raw data were plotted and statistically analyzed using GraphPad Prism 8 (GraphPad Software) unless indicated otherwise. Depending on the data set, ordinary one-way ANOVA (Dunnet's multiple-comparisons test) or two-way ANOVA (Tukey's or Sidak's multiple-comparisons test) were performed, and statistically significant differences were indicated by asterisks *<0.05, **<0.01, ***<0.001, ****<0.0001.

Affinities for binding to human FcRn at pH 6.0 determined using SPR and data from the cynomolgus monkey experiments were statistically analyzed using the "mixed" procedure in SAS Viya release V.03.04. The final models were run with restricted maximum likelihood. The assumptions of normality and homoscedasticity of the residuals with expectation zero were confirmed (not shown).

A linear model for the logarithm of the affinities was used with antibody as a fixed categorical effect. Heterogeneity in the residual variability between antibodies was incorporated in order to accommodate for differences in variability of the measured affinities to human FcRn for the different antibodies. Pairwise hypothesis tests were performed, and $P$ values were corrected by applying Hommel's procedure for multiple testing to have a family-wise error rate of 5%. For the cynomolgus monkey experiments, a longitudinal analysis model was built for the logarithm of the tracer antibody levels relative to predose to accommodate for any differences in tracer antibody levels between animals before the start of infusion. Fixed effects of treatment group, time in days post-infusion, and their interaction were included as independent categorical variables. A heterogeneous first-order autoregressive covariance structure was incorporated to allow heterogeneity of the variances at different time points and to accommodate for the correlation between repeated measurements within one animal. A joint hypothesis test was performed at a 5% significance level to evaluate if any difference between the longitudinal profiles of either anti-HEL IgG1-MST-HN or Fc-MST-HN with PBS could be detected. If significant, the individual hypotheses were tested and reported. Besides, also the longitudinal profiles of anti-HEL IgG1 and Fc treatments were directly compared and tested with α = 0.05. The denominator degrees of freedom for the hypothesis tests were approximated as detailed by Kenward and Roger[96].

## Reporting summary

Further information on research design is available in the Nature Research Reporting Summary linked to this article.

# Data availability

The datasets generated during and/or analyzed during the current study are available from the corresponding author upon reasonable request. Crystallographic coordinates and data are deposited in the PDB databank, under accession codes 7Q15 and 7Q3P. Source data are provided with this paper.

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

## Acknowledgements

The work of M.B. and E.v.d.K. was funded by argenx. E.P., J.A., and S.N.S. were supported by a grant from Flanders Innovation & Entrepreneurship (VLAIO), Belgium. We would like to thank Gerard van Mierlo, Pleuni Ooijevaar-de Heer, and Valerie Hanssens for technical assistance, Ariëlla van den Sompel for statistical analyses, Dr. Vladimir Bobkov for fruitful discussion, and Prof. E. Sally Ward for reading the manuscript. Savvas N. Savvides and Erwin Pannecoucke thank the staff of the EMBL beamlines at synchrotron PETRA3 (Hamburg, Germany) and the staff of Proxima2A at synchrotron SOLEIL (Saclay, France) for synchrotron beam time allocation and technical support.

## Author contributions

M.B., E.P., J.A., B.B., M.S., P.V., P.U., H.d.H., T.R., S.N.S., and G.V. designed the research. M.B., E.P., E.v.d.K., A.B., J.A., N.D., P.V., P.U., H.d.H., T.R., S.N.S., and G.V. designed the experiments. M.B., E.P., E.v.d.K., A.B., N.D., J.A., and B.B. performed the experiments. M.B., E.P., E.v.d.K., A.B., N.D., J.A., B.B., T.R., S.N.S., and G.V. analyzed the data. M.B., E.P., S.N.S., and G.V. wrote the manuscript. All authors contributed to and approved the manuscript.

## Competing interests

The authors declare the following competing interests: M.B., E.P., B.B., M.S., P.V., P.U., and H.d.H. are employed by argenx. G.V. serves as a consultant to argenx. The remaining authors declare no competing interests.

## Additional information

¹Immunoglobulin Research Laboratory, Department of Experimental Immunohematology, Sanquin Research and Landsteiner, Amsterdam UMC, University of Amsterdam, 1066 CX Amsterdam, The Netherlands. ²Department of Biomolecular Mass Spectrometry and Proteomics, Utrecht Institute for Pharmaceutical Sciences and Bijvoet Center for Biomolecular Research, Utrecht University, Utrecht, The Netherlands. ³argenx, 9052 Zwijnaarde, Belgium. ⁴Unit for Structural Biology, Department of Biochemistry and Microbiology, Ghent University, 9052 Ghent, Belgium. ⁵Unit for Structural Biology, VIB-UGent Center for Inflammation Research, 9052 Ghent, Belgium. ⁶Department of Immunopathology, Sanquin Research and Landsteiner Laboratory, Amsterdam UMC, University of Amsterdam, 1066 CX Amsterdam, The Netherlands. ✉e-mail: G.vidarsson@sanquin.nl

