## [Peer Review File · Nature Communications]

The Fab region of IgG impairs the internalization pathway of FcRn upon Fc engagementREVIEWER COMMENTS

Reviewer #1 (Remarks to the Author):

Brinkhaus et. al have prepared a well-designed and thoughtful study which demonstrates that the Fab arms on IgG molecules influence binding to FcRn only within the context of cell membrane. The authors present several lines of evidence to support their conclusion. A combination of X-ray structures, cellular assays and in vivo animal experiments are presented. Overall, I found the study to be well executed and scientifically sound. The combination of biochemical, in vitro and in vivo work together makes for a compelling and convincing study. The work is significant and has the potential to make an impact FcRn targeting therapies and immunotherapy in general.

Minor issues to be addressed:

1. Although overall the data is sounds and convincing, the findings are really based on an n=1. A single case of Fab arms influencing FcRn binding does not necessarily equate to a universal case. It may be that the amount of steric effect from the Fab arms will vary depending on the biophysical properties of the specific Fab arm.
2. The authors should report the kinetics values (k_a , k_d) from the SPR and the Chi² fit to the Langmuir model. This could be simply added to the supplement
3. FigS2C. Given the low resolution of one of the structures, conclusions based on the electron density of a single side chain are somewhat dubious. Perhaps the authors could include electron density in their figures to better illustrate the effect on Y252 they are observing.

Minor (grammatical) points:

P.7 Line 113 "pVITRO" not pVIRTO. Authors should probably also indicate which vector specifically.
P24 line 526 "FcRN-GFP express also..." should read "also express"
P28 lines 603and 610-611 in vitro should be in italics

Reviewer #2 (Remarks to the Author):

In this manuscript, Brinkhaus et al compared the biochemical, structural, and in vivo properties of a full-size IgG1-MST-HN to its Fc-only counterpart. These results imply a rationale for why Fc-only fragments are functionally superior to full-size IgG1 molecules in blocking IgG salvage in vivo, and are poised to influence therapeutic strategies targeting FcRn. There are several significant concerns that need to be addressed:

1. It seems that there are few novel findings in this study, but the authors overstated the arguments. For instance, many studies have already reported that full-size IgG1 binds FcRn indistinguishably from Fc-only IgG1. The authors solved the crystal structure of Fc-MST-HN complexed with FcRn, but still there was no evident difference with the previously reported Fc-MST. More importantly, unlike the native IgG1, the Fc only fragment as well as its variants were known to be more "sticky" and have significant non-specific binding. Therefore, more experiments have to be performed to support the claims that "Fc only binds membrane-associated FcRn more efficiently than full-size IgG1", and "IgG1-Fc-MST-HN shows higher occupancy of intracellular FcRn than full-size IgG1-MST-HN" – these may only be due to the sticky nature of isolated Fc.
2. Why did the authors choose anti-HEL IgG1s as the object of study to compare the intact antibody IgG and its Fc fragment, rather than other antibodies. More pairs of IgGs and various Fc variants should be studies to support the conclusion.
3. When calculating affinity, the author uses by a Langmuir 1:1 binding model, which is different from the crystallographic models given in Figure 2 (1 Fc combined with 2 FcRn). How to explain?
4. It has been debated whether the dimeric Fc binds to two molecules FcRn or only one and whether the dimeric state of Fc is required for efficient binding to FcRn. Several experiments showed a 2:1 FcRn/Fc binding stoichiometry, whereas the 1:1 FcRn-Fc complex was also observed

in some studies (PMID: 8478919, PMID: 8700168, PMID: 10504233, PMID: 10413524). I suggest the authors discuss more on these points and explain why they only showed 2:1 stoichiometry.

5. In this manuscript the authors claimed that IgG1-Fc-MST-HN has higher occupancy of intracellular FcRn, and more efficiency in blocking FcRn-dependent recycling than full-size IgG1-MST-HN. From these conclusions, how do the authors see the difference between the half-lives of IgG1-Fc-MST-HN and IgG1-MST-HN? The half-life in vivo between the two is also a key index for the evaluation of application.

Minor,

6. Line 522: Change "4°" to "4°C"

Reviewer #3 (Remarks to the Author):

Various biologics targeting FcRn are being developed for the treatment of autoimmune diseases. This therapeutic strategy is promising and attractive because it can reduce blood levels of pathogenic autoantibodies by inhibiting recycling of endogenous IgG by FcRn. In this study, an engineered Fc fragment (Fc-MST-HN) with enhanced binding to FcRn and its full-length IgG version have been investigated from various angles in terms of their binding mode to FcRn. The novel findings provided here are important for a better understanding of the mechanism of action of FcRn-targeted therapeutic agents and will be useful for the design of better molecules. They also provide fundamental information for improving the blood half-life of IgG base biologics.

In cell-based evaluation systems, the Fc fragment has been shown to have higher FcRn occupancy on cell surface and in intracellular vesicles than full-length IgG (Fig. 3, Fig. 4). The Fc fragment was shown to have a greater ability to inhibit IgG recycling in vitro, and it was also shown that the Fc fragment facilitates clearance of tracer antibody better than full-length IgG in monkeys (Fig. 5). This experiment in monkeys is valuable, as it reflects the clinical situation in human patients well.

Crystal structure analysis of Fc-MST-HN with FcRn has been performed, and based on its results, the hypothesis has been proposed; in the case of full-size IgG, its Fab region causes steric hindrance with the membrane, thus weakening its activity as an FcRn antagonist (Fig. 2). Although the proposal of this mechanism is considered to be the central value of this study, the hypothesis is not new. Crystallographic analysis of Fc-MST, a modified Fc similar to Fc-MST-HN, with FcRn had already been performed, and it was also inferred from other studies that the Fab region may cause steric hindrance with the cell membrane.

To show that the steric hindrance of the Fab region is the cause of worse FcRn antagonistic activity of full-size IgG, experiments have been performed using mutants with different numbers of Fabs (Fig. 3). However, it seems that this proof should have been done in a more carefully designed experiment. To show the effect of the number of Fab clearly, experiments should be performed with mutants using the same Fab but varying only its number. On the other hand, the comparison of Fc fragments with full-length IgG in FcRn occupancy and in vitro recycling should have been done using multiple different Fabs to show the generality of the phenomenon; it is to demonstrate that the phenomenon is not specific to the particular Fab (anti-HEL).

This study provides useful insights into FcRn-targeted therapeutic agents, but in order to be published in a high-quality journal such as Nature Communications, improvements in the experimental designs described above, as well as in the minor issues described below, are to be addressed.

Minor issues

Calling the Fc fragment as "IgG1-Fc-MST-HN" is confusing, as it can be confused with full-length IgG1. It should be called "Fc-MST-HN" to easily identify it as an Fc fragment.

583 IgG1-Fc-MST-HN reduced tracer antibody levels significantly more efficiently than IgG1-MST-HN or PBS control

This is very confusion, related to the inappropriate nomenclature described above.

In Fig. 3, the name "Control IgG1" is used, but it is not clear what this refers to.

It is not clear what "IgG1-IHH" is, nor how it differs from FcRn dead in Fig. 3.

There is a large discrepancy between the results of the experiment in Fig. 3 C and Fig. 3 D in the degree of reduction of anti-FcRn Fab (%) by IgG1-Fc-WT (at 500 nM). Why is this?

561 Furthermore, it may also affect the uptake process itself and/or lead to higher retention of the IgG1-Fc-MST-HN in the cells than its full-size IgG

This is over speculation. There is insufficient evidence to make such a claim.

Fig. 5 The symbols for PBS and anti-HEL IgG1-WT are similar and confusing.

If the Accompanying manuscript described in the Discussion is not accepted, the description of this needs to be corrected.

Response to reviewers

We thank the reviewers for their thorough and constructive evaluation of our manuscript and provide here a point-by-point reply to all comments/ suggestions brought forward. All changes in the main text are highlighted in yellow. Proposed deletions are shown with additional strikethroughs.

Reviewer #1 (Remarks to the Author):

Brinkhaus et. al have prepared a well-designed and thoughtful study which demonstrates that the Fab arms on IgG molecules influence binding to FcRn only within the context of cell membrane. The authors present several lines of evidence to support their conclusion. A combination of X-ray structures, cellular assays and in vivo animal experiments are presented. Overall, I found the study to be well executed and scientifically sound. The combination of biochemical, in vitro and in vivo work together makes for a compelling and convincing study. The work is significant and has the potential to make an impact FcRn targeting therapies and immunotherapy in general.

Minor issues to be addressed:

1. Although overall the data is sounds and convincing, the findings are really based on an $n=1$. A single case of Fab arms influencing FcRn binding does not necessarily equate to a universal case. It may be that the amount of steric effect from the Fab arms will vary depending on the biophysical properties of the specific Fab arm.

We thank the reviewer for this suggestion. To investigate if our finding is generalizable across IgGs with different specificities, we performed the FcRn occupancy assay with another two additional IgG1-MST-HN variants (anti-biotin & anti-IgE). The occupancy behavior of the two new full size IgG1-MST-HN overlapped with the initially described anti-HEL IgG1-MST-HN. We present these new data in the results section in line 209 to 212 and Supplementary Fig. 8: 'To test if this observation is of a general nature rather than Fab specific, we tested two other full size IgG1-MST-HN with different specificities. Receptor occupancy among all full size IgG1 overlapped and was significantly different from Fc-MST-HN (Supplementary Fig. 8).'

This further strengthens the other observations in our manuscript, which collectively support the hypothesis that the Fab acts as a steric component during FcRn-mediated recycling.

Please note that for these additional experiments summarized in Supplementary Figure 8, the anti-HEL IgG1-MST-HN and the Fc-MST-HN were included again, highlighting the reproducibility of the data and the robustness of the assay.

2. The authors should report the kinetics values (k_a , k_d) from the SPR and the Chi2 fit to the Langmuir model. This could be simply added to the supplement.

We thank the reviewer for this suggestion. Indeed, we should have been more clear about the applied analysis by SPR in the materials and methods section. We use an equilibrium analysis and fit a 1:1 Langmuir model to our SPR data, which is not based on on-rate and off-rate as independent measures. Given the multiplex biosensor format of the IBIS MX96 system, we spot various concentrations of the IgG molecules (ligand) and when injecting FcRn (analyte), we measure binding to all ligand concentrations in parallel. We fit the RU values at 360 sec (Fig. 1a), at the end of the association time where we assume that an equilibrium has been reached for each of the spots of the whole array

tested. We then calculate K_D values on the basis of an affinity plot with response at equilibrium over analyte concentration. To be transparent, we added such plots in the new supplementary figure 2 and referred to it in line 125 to 126 of the results section: 'The affinity plots used to calculate K_D values can be found in Supplementary Fig. 2.'

The K_D values we obtained compare well with data published elsewhere, as e.g. Vaccaro et al., *Nat Biot* (23), 2005; (Ref 28), where binding of anti-HEL antibodies to hFcRn was investigated independently in a Biacore system.

We apologize for having missed this and have added this information in line 517 of the materials and methods section, line 122 of the results section and line 1056 and 1198 of the corresponding figure legends. It now reads: '... K_D values derived by performing an equilibrium analysis and fitting a Langmuir 1:1 binding model...'

3. FigS2C. Given the low resolution of one of the structures, conclusions based on the electron density of a single side chain are somewhat dubious. Perhaps the authors could include electron density in their figures to better illustrate the effect on Y252 they are observing.

That is an excellent suggestion. We have now added 4 additional panels to Supplementary Figure 2 (Supplementary Fig. 2 D-G) showing the 2Fo-Fc electron density maps for each of the observed poses of the interaction involving Y252.

Minor (grammatical) points:

P.7 Line 113 "pVITRO" not pVIRTO. Authors should probably also indicate which vector specifically.

Thank you. We have corrected accordingly and added specific vector and manufacturer information (line 385).

P24 line 526 "FcRN-GFP express also..." should read "also express"

Thank you, we adapted accordingly (line 179).

P28 lines 603 and 610-611 *in vitro* should be in italics

Thank you for spotting this. We adapted accordingly (line 259 and line 266 to 267).

Reviewer #2 (Remarks to the Author):

In this manuscript, Brinkhaus et al compared the biochemical, structural, and *in vivo* properties of a full-size IgG1-MST-HN to its Fc-only counterpart. These results imply a rationale for why Fc-only fragments are functionally superior to full-size IgG1 molecules in blocking IgG salvage *in vivo*, and are poised to influence therapeutic strategies targeting FcRn. There are several significant concerns that need to be addressed:

1. It seems that there are few novel findings in this study, but the authors overstated the arguments. For instance, many studies have already reported that full-size IgG1 binds FcRn indistinguishably from Fc-only IgG1. The authors solved the crystal structure of Fc-MST-HN complexed with FcRn, but still there was no evident difference with the previously reported Fc-MST. More importantly, unlike the

native IgG1, the Fc only fragment as well as its variants were known to be more "sticky" and have significant non-specific binding. Therefore, more experiments have to be performed to support the claims that "Fc only binds membrane-associated FcRn more efficiently than full-size IgG1", and "IgG1-Fc-MST-HN shows higher occupancy of intracellular FcRn than full-size IgG1-MST-HN" – these may only be due to the sticky nature of isolated Fc.

We thank the reviewer for the suggestions. Regarding stickiness, we performed careful HPLC-SEC analyses of our full antibodies and fragments thereof, seeing no traces of stickiness (Supplementary Fig. 1 & 6). Furthermore, we did not see any signs of non-specific binding in FcRn affinity chromatography comparing both Fc only and full size IgG1 variants (Fig. 1c). Both of our cellular assays, the surface competition assay (used to investigate binding to membrane-associated FcRn) (Figure 3) as well as the intracellular FcRn occupancy assay (Figure 4), are based on an FcRn-specific competition readout. We use a labelled anti-FcRn Fab, which binds an epitope overlapping with the MST-HN binding site on FcRn, which allows us to investigate the binding of IgG to FcRn in a solely FcRn-dependent manner. The concentration-dependent effect with the MST-HN variants and a control not binding FcRn (IgG1-IHH) (Figure 3c), strongly suggests in our opinion that we are not reading out a non-specific signal due to general stickiness, but that the Fc only truly binds FcRn better in a specific manner. Also, we found an identical effect of varying the number of Fab-arms in WT IgG1 antibodies (as opposed to the higher affinity MST-HN variants) (Figure 3d), confirming that our readout is FcRn specific. Also in this assay, we show with another FcRn-dead variant (H435A) that our readout is FcRn specific. We added a small paragraph in the discussion, summarizing the points discussed above in line 300 to 304: 'No signs of non-specific binding of the Fc only were observed compared with full size IgG to neither of the column materials (Fig. 1c, Supplementary Fig. 1a, Supplementary Fig. 6). This taken together with the FcRn-specific nature of our competition-based cellular assays and the use of the FcRn-dead controls (Fig. 3 & Fig. 4) support the conclusion that the effect we observe is truly FcRn dependent.' We hope to have sufficiently addressed the concerns raised by providing this clarification.

2. Why did the authors choose anti-HEL IgG1s as the object of study to compare the intact antibody IgG and its Fc fragment, rather than other antibodies. More pairs of IgGs and various Fc variants should be studied to support the conclusion.

We thank the reviewers for the suggestion and we do agree that in order to generalize the finding, more specificities needed to be tested in the FcRn occupancy assay. We therefore performed the assay with two additional full size IgG1-MST-HN molecules with Fabs directed against two unrelated targets (anti-biotin & anti-IgE) and compared them to Fc-MST-HN and the anti-HEL IgG1-MST-HN (Supplementary Figure S8). The data overlaps with the initially tested anti-HEL IgG1-MST-HN. Supported by this newly provided data, we believe the phenomenon we describe is of a general nature rather than Fab specific. We refer to the data in the results section in line 209 to 212: 'To test if this observation is of a general nature rather than Fab specific, we tested two other full size IgG1-MST-HN with different specificities. Receptor occupancy among all full size IgG1 overlapped and was significantly different from Fc-MST-HN (Supplementary Fig. 8).'

Please note that for these additional experiments summarized in Supplementary Figure 8, the anti-HEL IgG1-MST-HN and the Fc-MST-HN were included again, highlighting the reproducibility of the data and the robustness of the assay.

The use of anti-HEL IgG as a model antibody dates to the initial publication by Vaccaro et al. who had reported the MST-HN mutated IgG Fc as a tool to modulate *in vivo* IgG levels [Vaccaro et al., *Nat Biot*

(23), 2005; (Ref 28). Neither *in vivo* off-target effects have been observed, nor have we ever observed non-specific binding to e.g. column material (Fig. 1c, Supplementary Fig. 1a, Supplementary Fig. 6). We therefore consider the anti-HEL antibody a 'neutral' antibody, which can serve as a model IgG, now verified with two additional antibodies (Supplementary Fig. 8).

3. When calculating affinity, the author uses by a Langmuir 1:1 binding model, which is different from the crystallographic models given in Figure 2 (1 Fc combined with 2 FcRn). How to explain?

In the SPR experiments described, Fc fragments were covalently immobilized to the chip and FcRn was flowed over the surface. As the 2 binding sites on the Fc fragment are considered to be independent (i.e. binding on one binding site is not expected to influence the binding properties of the second binding site), a Langmuir 1:1 binding can be applied. In contrast, when FcRn would be immobilized and the Fc fragment would be used as the analyte, a so-called 'bivalent analyte binding' model should be applied: binding at the second analyte binding site is facilitated by the fact that the first binding site already has brought the analyte and ligand in close proximity. Such avidity typically results in an underestimation of the dissociation rate constant and, consequently leads to a higher apparent affinity. As a result of these experimental considerations, it is generally recommended to capture bivalent antibodies and analyze them using a Langmuir 1:1 binding model, rather than applying them as an analyte in solution and attempt to deconvolute affinity and avidity (Abdiche et al., *mAbs* (7), 2015; (Ref 47). This rationale has been now briefly described in the Materials and method section (line 519 - 520). See also response to the next concern.

4. It has been debated whether the dimeric Fc binds to two molecules FcRn or only one and whether the dimeric state of Fc is required for efficient binding to FcRn. Several experiments showed a 2:1 FcRn/Fc binding stoichiometry, whereas the 1:1 FcRn-Fc complex was also observed in some studies (PMID: 8478919, PMID: 8700168, PMID: 10504233, PMID: 10413524). I suggest the authors discuss more on these points and explain why they only showed 2:1 stoichiometry.

In contrast to the solely technical considerations for measuring the affinities of IgG molecules to FcRn in an unbiased manner, it has been demonstrated in an *in vivo* context that both Fc halves are needed to exhibit full binding capacity and biological activity. This is why we think that showing and assuming a 2:1 binding stoichiometry is reasonable. We do appreciate the suggestion and added a paragraph to the discussion providing more details on these matters (line 310 to 312): 'Although both, 1:1 and 2:1 FcRn:IgG stoichiometries have been suggested (Ref 58-61), several groups have found that for full binding capacity and *in vivo* activity, both Fc halves are required (Ref 19,47,57,62,63), which is why we assume a 2:1 IgG:FcRn stoichiometry.'

5. In this manuscript the authors claimed that IgG1-Fc-MST-HN has higher occupancy of intracellular FcRn, and more efficiency in blocking FcRn-dependent recycling than full-size IgG1-MST-HN. From these conclusions, how do the authors see the difference between the half-lives of IgG1-Fc-MST-HN and IgG1-MST-HN? The half-life *in vivo* between the two is also a key index for the evaluation of application.

We thank the reviewer for the good question. The actual answer is extremely complicated; if one would factor in all variables, it is more complicated than serum half-life alone. In the discussion we hypothesize that the Fabs promote a more efficient release of full size IgGs compared to Fc only at the end of a recycling cycle when being exocytosed. We think that an Fc only – missing this repulsion effect

of the Fabs with the membrane – would have a shorter apparent serum half-life, but at the same time shows more efficient blocking as it binds to FcRn better than a full size IgG. However, this does not mean that it is not biologically active anymore, but rather that it cannot be detected in the serum anymore as it might stay bound to FcRn instead of being cleared. This is evident from the cited publication of the human phase 1 study: The Fc-MST-HN exhibits an apparent serum half-life of less than 4 days (SAD – 10 mg/ kg), but exhibits its maximum PD effect after 14 days (Ulrichts et al., *JCI* (128), 2018; (Ref 29). We now improved clarity in the text regarding half-life and added another paragraph to the discussion in line 360 to 364: ‘Accordingly, we hypothesize that full size IgGs have a longer apparent serum half-life compared to Fc only, as evident from Gjølborg et al., *submitted* and previous findings (Ref 63). However, in the context of antagonizing FcRn with an engineered Fc fragment, having Fabs as a steric entity would lead to less efficient blocking of FcRn. Furthermore, it seems that Fc-MST-HN remains bound to FcRn rather than actually being cleared. This would also explain...’

Minor,

6. Line 522: Change “4°” to “4°C”

Thank you for spotting this, it has been adapted accordingly (line 175).

Reviewer #3 (Remarks to the Author):

Various biologics targeting FcRn are being developed for the treatment of autoimmune diseases. This therapeutic strategy is promising and attractive because it can reduce blood levels of pathogenic autoantibodies by inhibiting recycling of endogenous IgG by FcRn.

In this study, an engineered Fc fragment (Fc-MST-HN) with enhanced binding to FcRn and its full-length IgG version have been investigated from various angles in terms of their binding mode to FcRn. The novel findings provided here are important for a better understanding of the mechanism of action of FcRn-targeted therapeutic agents and will be useful for the design of better molecules. They also provide fundamental information for improving the blood half-life of IgG base biologics.

In cell-based evaluation systems, the Fc fragment has been shown to have higher FcRn occupancy on cell surface and in intracellular vesicles than full-length IgG (Fig. 3, Fig. 4). The Fc fragment was shown to have a greater ability to inhibit IgG recycling in vitro, and it was also shown that the Fc fragment facilitates clearance of tracer antibody better than full-length IgG in monkeys (Fig. 5). This experiment in monkeys is valuable, as it reflects the clinical situation in human patients well.

Crystal structure analysis of Fc-MST-HN with FcRn has been performed, and based on its results, the hypothesis has been proposed; in the case of full-size IgG, its Fab region causes steric hindrance with the membrane, thus weakening its activity as an FcRn antagonist (Fig. 2). Although the proposal of this mechanism is considered to be the central value of this study, the hypothesis is not new. Crystallographic analysis of Fc-MST, a modified Fc similar to Fc-MST-HN, with FcRn had already been performed, and it was also inferred from other studies that the Fab region may cause steric hindrance with the cell membrane.

To show that the steric hindrance of the Fab region is the cause of worse FcRn antagonistic activity of full-size IgG, experiments have been performed using mutants with different numbers of Fabs (Fig. 3).

However, it seems that this proof should have been done in a more carefully designed experiment. To show the effect of the number of Fab clearly, experiments should be performed with mutants using the same Fab but varying only its number. On the other hand, the comparison of Fc fragments with full-length IgG in FcRn occupancy and in vitro recycling should have been done using multiple different Fabs to show the generality of the phenomenon; it is to demonstrate that the phenomenon is not specific to the particular Fab (anti-HEL).

We thank the reviewer for the valuable comments. It is true that also others have suggested that the Fab as a steric portion could have an influence on binding to FcRn as indicated in line 94 to 99 of the introduction as well as line 250 to 251 of the discussion. Nevertheless and to our knowledge, we are the first to report an experimental approach that allows direct assessment of IgG binding to membrane associated FcRn. In our surface competition assay with WT Fc backbones (Figure 3d), we demonstrated for an antibody with varying amount of Fabs against an irrelevant and non-disclosed target (Control IgG) that binding to membrane-associated FcRn negatively correlates with the amount of Fabs present on the Fc portion. Unfortunately, we could not perform the experiment using IgG1-MST-HN with a different number of Fabs as suggested by the reviewer, because this molecule was not available.

We do agree that to generalize the finding, more specificities needed to be tested in the FcRn occupancy assay. We therefore performed the assay with two more full size IgG1-MST-HN molecules with Fabs directed against two more irrelevant targets (anti-biotin & anti-IgE) and compared them to Fc-MST-HN and the anti-HEL IgG1-MST-HN (Supplementary Figure 8). The data largely overlaps with the initially tested anti-HEL IgG1-MST-HN. Supported by this newly provided data, we believe the phenomenon we describe is of a general nature rather than Fab specific. We refer to the data in the results section in line 209 to 212: 'To test if this observation is of a general nature rather than Fab specific, we tested two other full size IgG1-MST-HN with different specificities. Receptor occupancy among all full size IgG1 overlapped largely and was significantly different from Fc-MST-HN (Supplementary Fig. 8).'

Please note that for these additional experiments summarized in Supplementary Figure 8, the anti-HEL IgG1-MST-HN and the Fc-MST-HN were included again, highlighting the reproducibility of the data and the robustness of the assay.

This study provides useful insights into FcRn-targeted therapeutic agents, but in order to be published in a high-quality journal such as Nature Communications, improvements in the experimental designs described above, as well as in the minor issues described below, are to be addressed.

Minor issues

Calling the Fc fragment as "IgG1-Fc-MST-HN" is confusing, as it can be confused with full-length IgG1. It should be called "Fc-MST-HN" to easily identify it as an Fc fragment.

We agree and adopted this accordingly throughout the manuscript.

583 IgG1-Fc-MST-HN reduced tracer antibody levels significantly more efficiently than IgG1-MST-HN or PBS control

This is very confusion, related to the inappropriate nomenclature described above.

Please see above.

In Fig. 3, the name “Control IgG1” is used, but it is not clear what this refers to.

We agree that this was confusing. We have adapted the description of Control IgG1 from the materials and methods and also specified in the figure legend of figure 3 that Control IgG1 is an IgG1 with an irrelevant specificity against a non-disclosed target.

It now reads (line 1101): ‘Control IgG1 is an antibody directed against an irrelevant and non-disclosed target.’

It is not clear what “IgG1-IHH” is, nor how it differs from FcRn dead in Fig. 3.

We apologize for this. We have added more information and references on IgG1-IHH in the materials and methods (line 412 to 413): ‘Linear DNA strands encoding for IgG1-WT and IHH (I253A, H310A, H435A: a triple AA substitution in IgG1 known to abrogate FcRn binding (Ref 48, 70)’ and in the results section (line 119 to 120): ‘In line with previously published data, no binding was observed for human IgG1-IHH, regardless of the pH assayed (Ref 48).’

There is a large discrepancy between the results of the experiment in Fig. 3 C and Fig. 3 D in the degree of reduction of anti-FcRn Fab (%) by IgG1-Fc-WT (at 500 nM). Why is this?

We thank the reviewer for the valid question. We specify in the materials and methods section as well as in the figure legends that the molar ratios of detection:inhibitor ratios are different between the setup in which we measure the MST-HN variants (Figure 3c) [ratio 1:25] and the WT variants (Figure 3d) [ratio 1:250]. We need to lower the concentration of the labelled anti-FcRn Fab (detection agent) when measuring the WT variants to compensate for the lower affinity of the WT antibodies compared to the MST-HN antibodies. The ratios used are a results of initial optimization experiments in which both, inhibitor (MST-HN and WT) and anti-FcRn Fab, have been titrated against each other to pinpoint the optimal detection window, while still being technically feasible (data not shown).

We do agree that this might have been hard to grasp and apologize for the confusion in the text. We specified the differences in a clearer manner in the figure legend of figure 3.

It now reads (line 1095 to 1097): Detection (AF650-labeled anti-FcRn Fab):inhibitor molar ratio at highest inhibitor concentration = 1:25 (optimized to measure competition with MST-HN containing IgG1) (...) (and line 1101 to 1103): Detection (AF650-labeled anti-FcRn Fab):inhibitor molar ratio at highest inhibitor concentration = 1:250 (optimized to measure competition with WT IgG).

561 Furthermore, it may also affect the uptake process itself and/or lead to higher retention of the IgG1-Fc-MST-HN in the cells than its full-size IgG

This is over speculation. There is insufficient evidence to make such a claim.

We agree with this reviewer and have now removed 'affect the uptake process itself and/or lead to' from this sentence. It now reads (line 217 to 219): 'Furthermore, it may also lead to higher retention of the Fc-MST-HN in the cells than its full-size IgG counterpart.'

Fig. 5 The symbols for PBS and anti-HEL IgG1-WT are similar and confusing.

We agree and have adapted the symbol of the PBS group to a more distinguishable one.

If the Accompanying manuscript described in the Discussion is not accepted, the description of this needs to be corrected.

We agree. Unfortunately, while the accompanying manuscript has been declined at Nature Communications, it was transferred to another journal within the Nature publishing group. We would like to keep the reference to this accompanying manuscript, given the complementary character of the described findings, unless of course, significant delays occur.

REVIEWERS' COMMENTS

Reviewer #1 (Remarks to the Author):

The authors have adequately addressed all the reviewer comments. I am satisfied and believe the manuscript is ready for publication.

Reviewer #2 (Remarks to the Author):

All my previous concerns have been addressed.

Reviewer #3 (Remarks to the Author):

One of the central values of this study is that steric hindrance of the Fab region disrupts binding to FcRn, and new experiments using Fabs with two different specificities were added in this regard. The same results were obtained for these Fabs, reinforcing the authors' argument and strongly suggesting that this mechanism is general.

With this additional experiment performed, the major concerns I had were addressed.

All of the other minor points raised have been properly corrected, and I believe that the manuscript is now significantly improved and more understandable to the reader.

In light of the above sincere and appropriate responses, I agree to publish this article in nature communication.

Response to the reviewers

Reviewer #1 (Remarks to the Author):

The authors have adequately addressed all the reviewer comments. I am satisfied and believe the manuscript is ready for publication.

Reviewer #2 (Remarks to the Author):

All my previous concerns have been addressed.

Reviewer #3 (Remarks to the Author):

One of the central values of this study is that steric hindrance of the Fab region disrupts binding to FcRn, and new experiments using Fabs with two different specificities were added in this regard.

The same results were obtained for these Fabs, reinforcing the authors' argument and strongly suggesting that this mechanism is general.

With this additional experiment performed, the major concerns I had were addressed.

All of the other minor points raised have been properly corrected, and I believe that the manuscript is now significantly improved and more understandable to the reader.

In light of the above sincere and appropriate responses, I agree to publish this article in nature communication.

Response from the author:

As there were no further points for discussion, we have not addressed any of the reviewers answers.